# Parthenolide disrupts mitosis by inhibiting ZNF207/BUGZ-promoted kinetochore-microtubule attachment

Susana Eibes[1,6], R Bhagya Lakshmi[1,6], Girish Rajendraprasad [1], Brian T Weinert[2], Fadhil S Kamounah[3], Luke F Gamon[4], Sergi Rodriguez-Calado [1], Morten Meldal [3], Michael J Davies [4], Michael Pittelkow[3], Chunaram Choudhary [2] & Marin Barisic [1,5]✉

## Abstract

**Parthenolide is a natural compound that has shown highly promising anticancer activity. Even though its mode of action has been studied for decades, its antimitotic activity has been largely overlooked, limiting the understanding of its full anticancer potential. In this study, we combined click-chemistry with quantitative mass spectrometry and cell biology to elucidate the mechanism of action of parthenolide in mitosis. We show that parthenolide does not act as a microtubule-targeting agent in cells. Instead, it binds to the kinetochore protein ZNF207/BUGZ, preventing the establishment of proper kinetochore-microtubule attachment. Our results show that parthenolide covalently binds to Cys54 of BUGZ via Michael addition to its α-methylene-γ-lactone moiety. Since Cys54 is located within the second zinc-finger domain of the BUGZ microtubule-targeting region, we propose that parthenolide interferes with the microtubule-binding ability of BUGZ, consequently preventing kinetochore-microtubule attachments required for accurate chromosome congression to the spindle equator.**

**Keywords** BUGZ; Kinetochore; Microtubules; Mitosis; Parthenolide
**Subject Categories** Cell Adhesion, Polarity & Cytoskeleton; Cell Cycle; Pharmacology & Drug Discovery

## Introduction

Parthenolide (PTL) is a natural compound extracted from feverfew (*Tanacetum parthenium*), a plant that has been used for centuries for various medicinal purposes (Pareek et al, 2011). By chemical structure, it is a sesquiterpene lactone that contains an electrophilic α-methylene-γ-lactone moiety responsible for PTL's reactivity with nucleophiles, particularly thiols. The interaction between its electrophilic α-methylene-γ-lactone group and the nucleophilic thiol group present in cysteines of the target proteins involves a Michael addition reaction, during which the nucleophile attacks the electrophile, thereby forming a covalent bond (Freund et al, 2020).

PTL is mainly known for its anti-inflammatory and anticancer effects (Carlisi et al, 2022; Ghantous et al, 2013; Sztiller-Sikorska and Czyz, 2020). Its high potential as an anti-cancer agent lies in its ability to selectively induce programmed death of cancer cells, as well as in its efficiency in targeting cancer stem cells that can initiate tumor formation, causing relapses and resistance against chemotherapy (Carlisi et al, 2022; Ghantous et al, 2013; Sztiller-Sikorska and Czyz, 2020). Although its precise anticancer mode of action is still being studied, PTL affects several cancer-associated molecular pathways. For instance, it was shown to induce apoptosis by inhibiting prosurvival transcription factors NF-κB (Hehner et al, 1999; Kwok et al, 2001; Saadane et al, 2007) and STAT proteins (Carlisi et al, 2011; Legendre et al, 2003; Sobota et al, 2000), as well as by inducing oxidative stress via promoting an increase in reactive oxygen species (ROS) (Wang et al, 2006; Wen et al, 2002; Zhang et al, 2004). In addition, we and others have previously demonstrated that PTL disrupts accurate chromosome congression during cell division, resulting in severe mitotic arrest (Barisic et al, 2015; Hotta et al, 2021).

Although the impact of PTL on mitosis may be a crucial aspect of its anticancer activity, the underlying molecular mechanism remains largely unknown. Initially, the effect of PTL on mitosis was attributed to its ability to hinder microtubule detyrosination (Fonrose et al, 2007) that guides kinesin-7/CENP-E-mediated chromosome transport to the cell equator (Barisic and Maiato, 2016; Barisic et al, 2015). However, a recent study showed that PTL covalently binds tubulin in vitro, suggesting that its impact on mitosis may arise from direct tubulin binding (Hotta et al, 2021). Interestingly, PTL and its analogs were shown to sensitize cancer cells to the effects of microtubule-targeting agents frequently used in chemotherapy, such as taxanes and vinca alkaloids

[1]Cell Division and Cytoskeleton, Danish Cancer Institute, Copenhagen, Denmark. [2]Department of Proteomics, The Novo Nordisk Foundation Center for Protein Research, Faculty of Health and Medical Sciences, University of Copenhagen, Copenhagen, Denmark. [3]Department of Chemistry, University of Copenhagen, Copenhagen, Denmark. [4]Department of Biomedical Sciences, Panum Institute, University of Copenhagen, Copenhagen, Denmark. [5]Department of Cellular and Molecular Medicine, Faculty of Health and Medical Sciences, University of Copenhagen, Copenhagen, Denmark. [6]These authors contributed equally: Susana Eibes, Bhagya Lakshmi R. ✉E-mail: barisic@cancer.dk

(Liu et al, 2008; Shanmugam et al, 2006; Shanmugam et al, 2010; Sweeney et al, 2005; Zhang et al, 2009).

To elucidate the mechanism of action of PTL in mitosis and identify its mitotic target(s), we designed and synthesized a PTL-based click chemistry probe, which we used for fluorescence imaging and quantitative affinity purification-mass spectrometry. We show that, in cellulo, PTL targets several spindle and kinetochore proteins rather than tubulin. Moreover, PTL does not disrupt microtubule dynamics during interphase and mitosis. Instead, it localizes at kinetochores, impeding the formation of stable kinetochore-microtubule attachments. We identified ZNF207/BUGZ as the primary mitotic target of PTL and propose the mode of action of PTL in mitosis.

## Results

### Parthenolide does not interfere with microtubule dynamics in cells

PTL induces severe problems during mitosis, consequently preventing cell division and causing a cell cycle arrest. This strong phenotype, described in two independent studies (Barisic et al, 2015; Hotta et al, 2021), has been generally overlooked for decades and has not been considered in the context of PTL mechanism of action.

To better understand the impact of PTL on dividing cells, we performed live-cell imaging of human osteosarcoma U2OS cells stably expressing H2B-GFP and mScarlet-α-tubulin, treated with DMSO or 15 μM PTL 10 min before filming. The cells that entered mitosis displayed severe problems, demonstrating a rapid effect of this drug on the accuracy of cell division. Although PTL-treated cells did not present any obvious defect in microtubule nucleation and polymerization in early mitosis, the cells failed to assemble a robust mitotic spindle and remained arrested in mitosis, displaying severe problems in chromosome congression (Fig. 1A; Movie EV1). Since tubulin is a frequent off-target in small molecule-based drug discovery programs, we tested whether the major impact of PTL on mitotic spindles arises from its direct binding to tubulin that disrupts microtubule dynamics. To test the effect of the compound on microtubules in vitro, we performed a turbidity-based microtubule polymerization assay in the presence of PTL. As previously reported (Hotta et al, 2021), PTL indeed slowed down microtubule polymerization in this assay (Appendix Fig. S1A). However, since in vitro analyses do not reflect the cellular context, we next analyzed the effect of PTL on microtubule dynamics in cells. To do so, we performed live-cell imaging of U2OS cells stably expressing GFP-tagged EB1, a microtubule plus-end tracking protein that flags the tips of growing microtubules. By tracking the EB1-GFP comets in interphase cells, we demonstrated that PTL does not interfere with microtubule growth in cellulo, as represented by the EB1-GFP traveled distance (13.71 ± 2.36 μm in DMSO; 13.85 ± 2.81 μm in PTL), lifetime (18.42 ± 2.29 s in DMSO; 18.51 ± 2.04 s in PTL) and velocity (0.75 ± 0.11 μm/s in DMSO; 0.75 ± 0.13 μm/s in PTL) (Fig. 1B,C; Appendix S1B). Moreover, we performed a microtubule regrowth assay in U2OS cells pre-treated with either DMSO or PTL for 2 h prior to a cold shock. Microtubule nucleation was not affected by PTL treatment, as indicated by tubulin intensity at the centrosomes measured 1 and 5 min after releasing the cells from the

cold shock (t = 1 min: 1.1 ± 0.62 in DMSO and 1.37 ± 0.95 in PTL; t = 5 min: 1.08 ± 0.36 in DMSO and 1.1 ± 0.39 in PTL). The microtubule network was completely re-established 5 min after the cold shock both in control and PTL-treated cells (Fig. 1D,E). Taking these results together, we conclude that PTL does not interfere with microtubule polymerization nor nucleation in interphase cells.

Mitosis is a very dynamic process in which a robust structure, the mitotic spindle, must be assembled and disassembled within a short period of time to promote chromosome alignment and accurate cell division. Therefore, microtubules in mitosis are much more dynamic than during interphase (Rusan et al, 2001) and PTL may have a selective effect on microtubule dynamics during mitosis. To test this, we analyzed the effect of PTL on the dynamics of mitotic spindle microtubules. To prevent potential impact of uncongressed polar chromosomes on the measurement of microtubule dynamics, we used the anaphase promoting complex (APC) inhibitors Apcin and Protame to arrest U2OS cells in metaphase prior to adding PTL. Our previous work showed how microtubule-targeting agents interfere with the most dynamic microtubules of mitotic spindle, the astral microtubules, in metaphase cells (Rajendraprasad et al, 2021; Vit et al, 2022). In contrast, PTL did not affect the appearance of astral microtubules, which displayed intensities comparable to control cells (1.05 ± 0.48 in DMSO; 1.1 ± 0.44 in PTL). Moreover, the length of mitotic spindle was also not altered (13.13 ± 1.4 μm in DMSO; 13.35 ± 1.5 μm in PTL) (Fig. 1F,G). In addition to observing unaltered physical appearance of the mitotic spindle, we also performed a more detailed analysis of microtubule dynamics in mitosis. We measured fluorescence dissipation after photoactivation to quantify microtubule turnover in metaphase-arrested U2OS cells stably expressing photo-activatable (PA)-GFP and mCherry-α-tubulin. Our results revealed that PTL does not affect microtubule turnover in metaphase spindles (Fig. 1H,I) (microtubule half-life: Slow fraction – 183.4 ± 17.56 s in DMSO; 176.9 ± 10.91 s in PTL, Fast fraction – 14.49 ± 0.34 s in DMSO; 15.25 ± 1.69 s in PTL).

### Parthenolide obstructs kinetochore-microtubule attachment

Although microtubule dynamics were not affected upon the addition of PTL in metaphase (Fig. 1F–I), PTL interfered with the formation of a robust mitotic spindle when it was added to cells prior to mitosis (Fig. 1A). This indicates that PTL impedes spindle assembly without directly binding to microtubules. To dissect the effect of this compound in spindle assembly, we monitored microtubule regrowth after a cold shock in mitotic cells. Live-cell imaging of PTL-treated U2OS cells showed that PTL does not interfere with microtubule nucleation and growth associated with the initial stages of spindle assembly, since tubulin intensity at centrosomes was equal in DMSO- and PTL-treated cells 1 min after a release from the cold shock (1.06 ± 0.51 in PTL, compared to 1.1 ± 0.43 in DMSO). However, during the later stages of spindle assembly (e.g., 20 min after the release from the cold shock) PTL prevented the formation of a functional spindle and induced problems with chromosome congression (Fig. 2A,B). This suggests that PTL interferes with the formation of stable bundles of microtubules that connect to kinetochores (also known as k-fibers) rather than with microtubule dynamics in general.

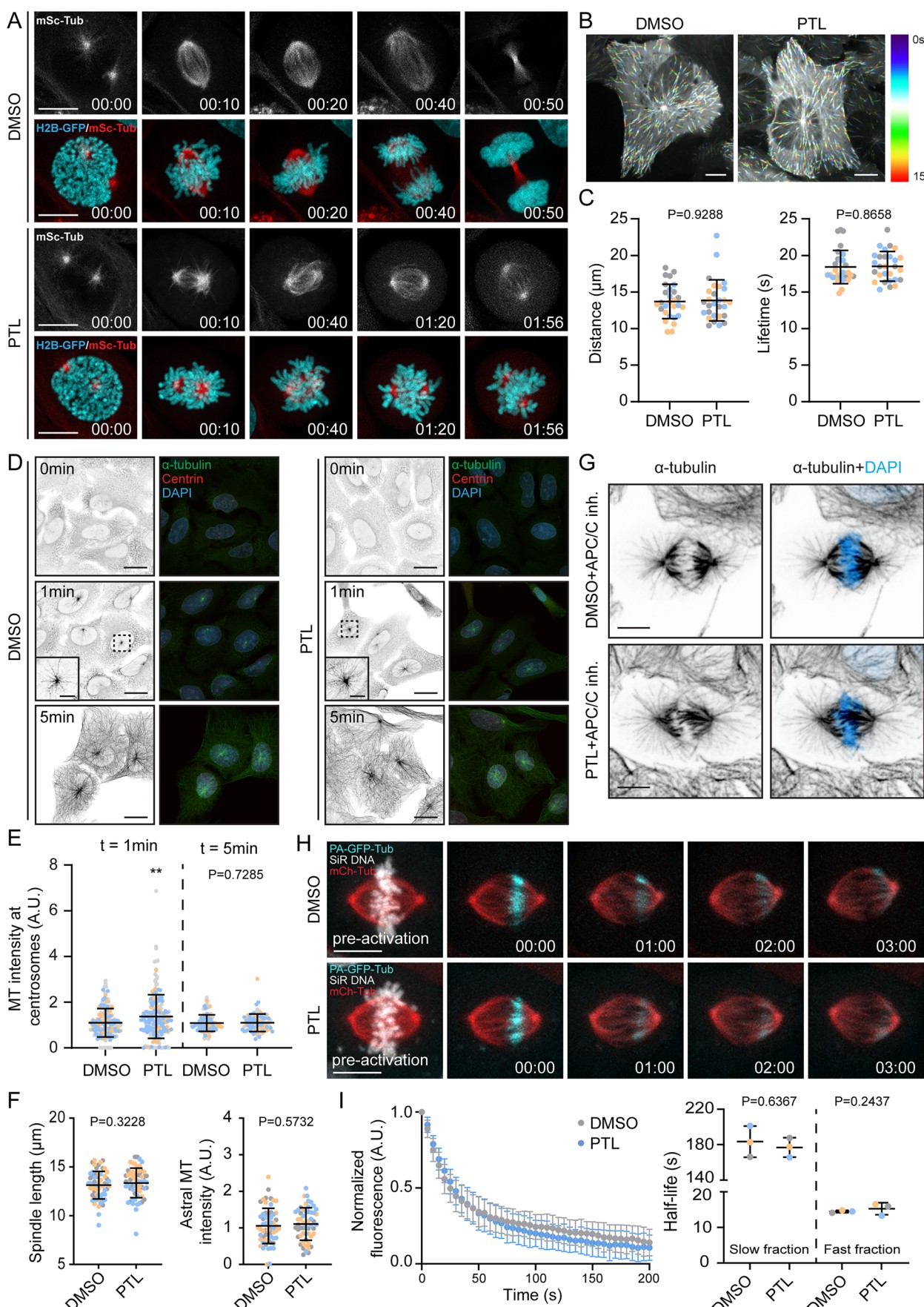

◄ **Figure 1. Parthenolide does not interfere with microtubule dynamics in cells.**

(A) Representative spinning-disk confocal time-series of mitotic U2OS H2B-GFP/mScarlet-α-tubulin cells treated with DMSO or 15 μM PTL. Time: hour:min. Scale bar: 10 μm. (B) Representative color-coded temporal projections of U2OS EB1-GFP control or 15 μM PTL treated cells. Scale bar: 10 μm. (C) Quantification of distance traveled and lifetime by EB1-GFP comets with the indicated treatments. N, n (N = number of cells, n = number of experiments) DMSO (28, 3), 15 μM PTL (28, 3). (D) Representative confocal max projections of different time points of microtubule regrowth of DMSO- and 15 μM PTL-treated U2OS cells. Scale bar: 10 μm. Zoomed areas represent centrosomal microtubule nucleation in time point = 1 min. (E) Quantification of tubulin intensity at centrosomes at time points = 1 and 5 min. N, n (N = number of cells, n = number of experiments): DMSO (155, 3), 15 μM PTL (196, 3). **p ≤ 0.01. (F) Quantification of spindle length and astral microtubule intensity in metaphase arrested cells treated with DMSO or 15 μM PTL. N, n (number of cells, number of experiments) for spindle length: DMSO (60, 3), 15 μM PTL (59, 3). N, n (N = number of cells, n = number of experiments) for astral intensity: DMSO (60, 3), 15 μM PTL (54, 3). (G) Representative spinning-disk confocal max projections of metaphase U2OS cells arrested with Apcin and proTAME undergoing the indicated treatments. Scale bar: 10 μm. (H) Representative spinning-disk confocal time-lapse images showing microtubule turnover in U2OS-PA-GFP/mCherry-α-tubulin cells undergoing indicated treatments. Scale bar: 10 μm. (I) Exponential decay of normalized fluorescence intensity of photoactivated α-tubulin over time and microtubule half-lives. N, n (number of cells, number of experiments): DMSO (29, 3), 15 μM PTL (30, 3). Half-lives calculated from the decay curves of three independent experiments are plotted. Replicates are color-coded for all quantifications. All data are presented as mean ± SD values from three independent replicates. Statistical analysis was performed using unpaired t-test for the analysis of EB1 lifetime in (C), astral microtubule intensity in (F) and microtubule half-lives in (I). Mann-Whitney test was used for EB1 distance in (C), data in (E) and spindle length in (F). Source data are available online for this figure.

To study the effect of PTL on k-fiber formation, we performed a classical cold-treatment assay, in which U2OS cells were subjected to a short cold treatment that depolymerizes dynamic microtubules, while stable and bundled microtubules are kept intact. Tubulin intensity at the mitotic spindle of PTL-treated cells was significantly reduced after the cold treatment (0.53 ± 0.3 in PTL, compared to 1 ± 0.27 in DMSO), suggesting that the cells treated with PTL cannot form k-fibers properly (Fig. 2C,D).

Since the formation of k-fibers is characterized by stable end-on kinetochore-microtubule attachments, we analyzed the attachment status of kinetochores in PTL-treated cells. Astrin is a microtubule associated protein that plays a key role in maintaining and stabilizing the kinetochore-microtubule attachments. It is recruited to kinetochores only after chromosome bi-orientation, when both kinetochores are end-on attached to the microtubules emanating from the opposite spindle poles. Therefore, astrin can be used as a marker of stable kinetochore-microtubule attachments (Shrestha and Draviam, 2013). To analyze the status of kinetochore-microtubule attachment in U2OS cells treated with PTL, we performed immunofluorescence against astrin, using CENP-C as a kinetochore reference marker. Astrin intensity at congressed kinetochores was strongly reduced in PTL-treated cells (0.45 ± 0.3 in PTL, compared to 1 ± 0.27 in DMSO) (Fig. 2E,F). Altogether, our results demonstrate that PTL prevents the formation of stable kinetochore-microtubule attachment, thereby disrupting the assembly of mitotic spindle and chromosome alignment.

## Parthenolide binds to mitotic spindle and kinetochores in mitosis

Although the above data indicates that PTL interferes with the formation of stable kinetochore-microtubule attachments during mitosis, without directly targeting the microtubules, the target of PTL that could explain this cellular phenotype remains unknown. In order to identify mitotic target(s) of PTL, we synthesized a clickable PTL derivative, hereafter referred to as alkyne-PTL, containing a polyethylene glycol linker with a terminal alkyne group (Fig. EV1A). Functionalization was done on the allylic methyl (C11) group by SeO₂ mediated oxidation to the allylic alcohol (Ge et al, 2019), followed by installation of the terminal alkyne through a PEGylation reaction. This modification does not interfere with any of the putative reactive sites of PTL, namely the

epoxide and the α-methylene-γ-lactone. This synthetic strategy allows the electrophilic sites in the PTL structure to remain intact. Both functionalities provide an electrophilic center for covalent binding with nucleophilic Cys residues from targeted proteins. However, the epoxide site is not required for the mitotic phenotype, given that PTL analog costunolide, which lacks this group, is an active molecule that phenocopies the described mitotic problems (Figs. 3A and EV1B–E). These results suggest that PTL covalently binds mitotic target(s) via Michael addition of a Cys residue to the α-methylene-γ-lactone.

To validate the activity of alkyne-PTL, we treated U2OS cells for 2 h with the modified compound and imaged mitotic cells. Similar as in PTL- and costunolide-treated cells, tubulin intensity at the mitotic spindle in cells treated with alkyne-PTL was significantly reduced after the cold treatment (0.51 ± 0.26 in alkyne-PTL and 0.28 ± 0.20 in costunolide, compared to 1.03 ± 0.45 in DMSO) (Fig. EV1B,C). Moreover, astrin intensity at congressed kinetochores was also strongly reduced in alkyne-PTL-treated cells (0.75 ± 0.47 in alkyne-PTL and 0.62 ± 0.25 in costunolide, compared to 1.12 ± 0.54 in DMSO) (Fig. EV1D,E). Furthermore, by measuring the relative intensity of detyrosinated tubulin by immunoblotting, we showed that alkyne-PTL retained the inhibiting effect of PTL on tubulin detyrosination (0.2 ± 0.17 in PTL; 0.45 ± 0.03 in alkyne-PTL; 0.58 ± 0.1 in costunolide; compared to 1 ± 0 in DMSO) (Fig. EV1F). Altogether, our results show that alkyne-PTL is an active molecule that preserved the abilities to induce mitotic problems and reduce the amount of detyrosinated microtubules similar to unmodified PTL (Figs. 3A and EV1B–F).

In order to disclose the location of the protein(s) targeted by PTL, we performed a Copper-catalyzed Azide-Alkyne Cycloaddition (CuAAC) click reaction (Rostovtsev et al, 2002; Tornoe et al, 2002) with an azide-fluorophore in fixed cells previously treated with 15 μM alkyne-PTL as represented in Appendix Fig. S2A. Alkyne-PTL displayed a predominantly nuclear localization during interphase, with a small fraction being co-localized with actin stress-fibers. In line with our results showing no effect of PTL on microtubule dynamics, alkyne-PTL did not co-localize with microtubules, further confirming that PTL is not a microtubule-targeting agent (Fig. 3B). Importantly, the localization of alkyne-PTL drastically changed during cell division. Its nuclear localization vanished after nuclear envelope breakdown, indicating that PTL does not target a chromatin-associated protein. In

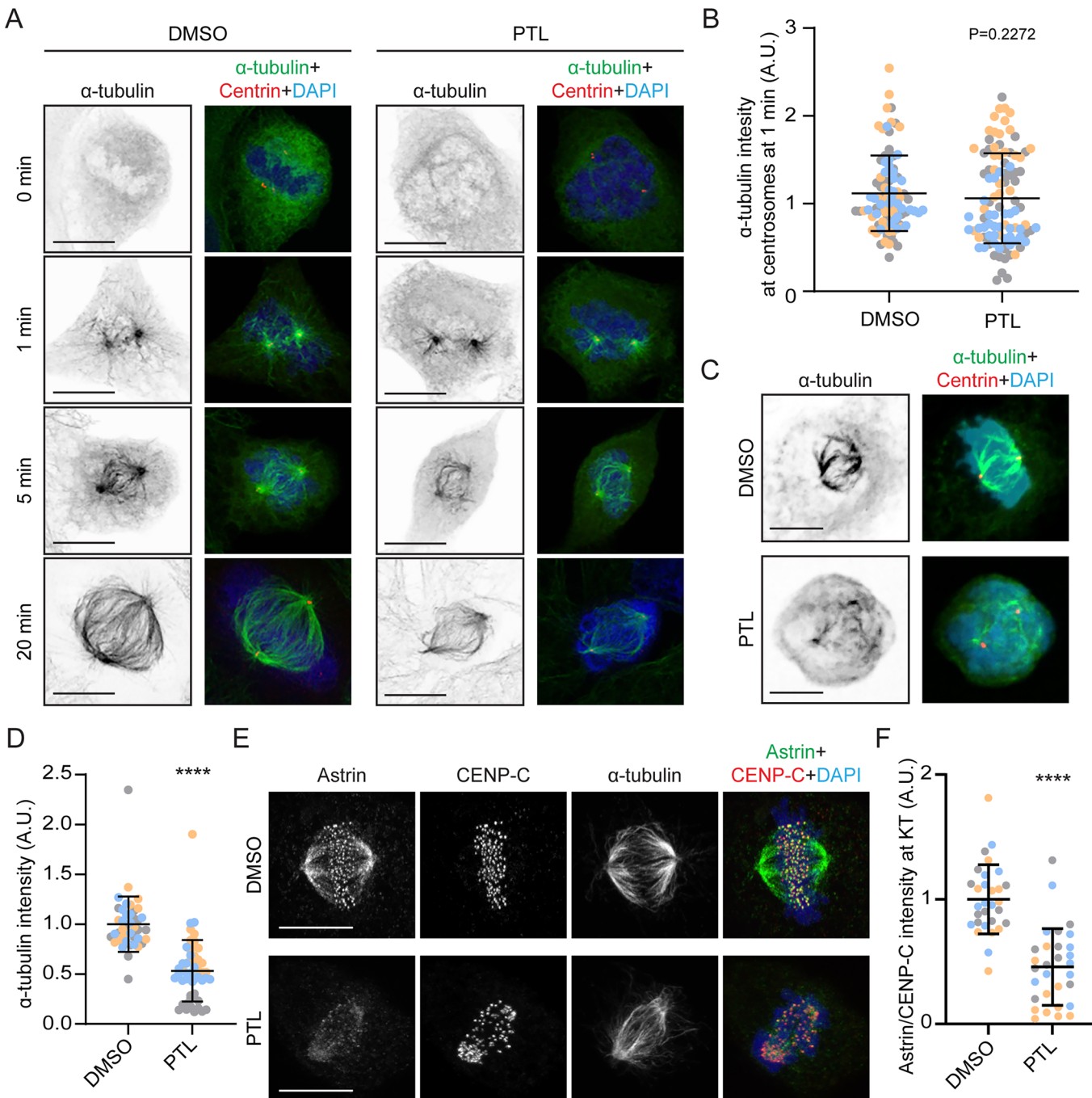

**Figure 2. Parthenolide interferes with stable kinetochore-microtubule attachments.**

(A) Representative confocal max projections of different time points of microtubule regrowth of U2OS cells treated with DMSO or 15 μM PTL. Scale bar: 10 μm. (B) Quantification of α-tubulin intensity at centrosomes at time point = 1 min. N, n (N = number of cells, n = number of experiments) DMSO (92, 3), 15 μM PTL (97, 3). (C) Representative spinning-disk confocal max projections of DMSO- and 15 μM PTL-treated U2OS cells subjected to a short cold treatment. Scale bar 10 μm. (D) Quantification of α-tubulin intensity at the mitotic spindle of cells treated as indicated in (C). N, n (N = number of cells, n = number of experiments): DMSO (42, 3), 15 μM PTL (52, 3). ****p ≤ 0.0001. (E) Confocal max projections of mitotic U2OS cells undergoing the indicated treatments immunostained with indicated antibodies and DAPI as DNA counter stain. Scale bar: 10 μm. (F) Quantification of astrin intensity at aligned kinetochores normalized to CENP-C intensity for DMSO and 15 μM PTL treated cells. N, n (N = number of cells, n = number of experiments): DMSO (30, 3), 15 μM PTL (30, 3). ****p ≤ 0.0001. Each color represents an individual experiment for all quantifications. All data are presented as mean ± SD values from three independent replicates. Statistical analysis was performed using Mann-Whitney test for (B) and (D) and unpaired t-test for (F). Source data are available online for this figure.

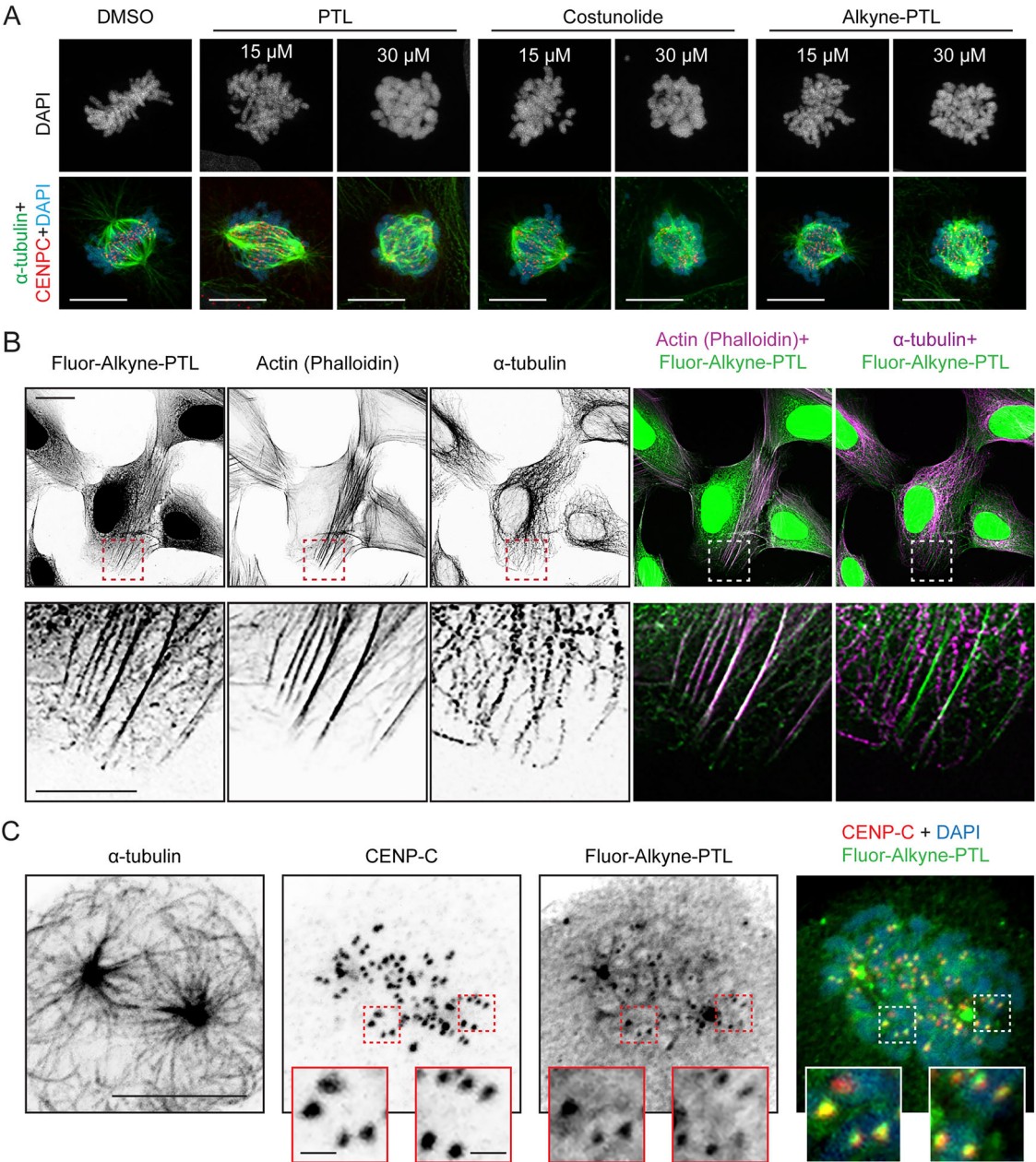

**Figure 3. Parthenolide targets kinetochore and spindle-associated proteins.**

(A) Representative confocal max projections of dividing U2OS cells undergoing the stipulated treatments and concentrations. Scale bar: 10 μm. (B) Representative confocal maximum projections of click-based imaging of 15 μM fluor-488-alkyne-PTL-treated interphase cells co-stained with α-tubulin and phallodin, as markers for microtubules and actin fibers, respectively. Zoomed areas show co-localization with actin fibers. Scale bar: 20 μm. Zoomed inset scale bar: 10 μm. (C) Representative confocal max projections of click-based imaging of 15 μM fluor-488-alkyne-PTL-treated prometaphase U2OS cells co-stained with α-tubulin and CENP-C, as markers for microtubules and kinetochores, respectively. Zoomed cropped areas show co-localization with kinetochores. Scale bar: 10 μm. Zoomed inset scale bar: 1 μm. Source data are available online for this figure.

prometaphase, alkyne-PTL localized to kinetochores and mitotic spindle (Fig. 3C; Appendix Fig. S2B). Using lower concentrations of alkyne-PTL (5 μM), we could overcome the mitotic arrest and monitor the compound localization throughout mitosis. Alkyne-PTL displayed a weaker kinetochore localization during later stages of mitosis compared to early prometaphase, indicating that

the targeted protein is removed from kinetochores upon the establishment of stable kinetochore-microtubule attachments (Appendix Fig. S2B).

These results suggest that PTL interference with kinetochore-microtubule attachment is a consequence of direct targeting of a mitotic spindle- and/or a kinetochore-associated protein(s).

## BUGZ is the main mitotic target of parthenolide

In order to identify the targets of PTL that are linked to its antimitotic activity, we combined a click-based pull-down of the molecule with quantitative liquid chromatography-mass spectrometry (LC-MS). To validate the target identification, we performed a competition assay in which U2OS cells were treated with or without an excess of free PTL prior to the incubation with alkyne-PTL. Taking advantage of the covalent nature of the compound, we first performed target labeling by clicking alkyne-PTL to 488-fluor-azide in lysates from previously treated cells. In-gel fluorescence confirms the expected reactivity of the natural compound, which targets numerous proteins (Appendix Fig. S3A). After validation, we performed a competition pull-down by clicking alkyne-PTL to azide agarose beads for quantitative LC-MS (proteomic) analysis. To do so, we treated SILAC-labeled cells with alkyne-PTL with or without an excess of free PTL to promote competition. After the click reaction, proteins bound to PTL and, in consequence, to the agarose beads were digested and analyzed by LC-MS (Fig. 4A). PTL-binding proteins were filtered by selectivity (SILAC H/L ratio higher than 2) and reproducibility (H/L abundances $p$-value lower than 0.05). The pull-downs confirmed the reactivity of the compound, disclosing 182 proteins that fulfilled the selected requirements to be considered as targets (Fig. 4A). Correlations between replicates were high, indicating the efficacy of the procedure (Appendix Fig. S3B). Next, we performed gene ontology (GO) enrichment analysis to gain mechanistic insights (Thomas et al, 2022). PTL-targeted proteins are mainly nuclear and associated with the actin cytoskeleton. Furthermore, some of the strongest targets belong to redox pathways, specifically to the glutathione and thioredoxin antioxidant pathways (Appendix Fig. S3C), confirming previous publications (Wang et al, 2006; Wen et al, 2002; Zhang et al, 2004) and validating the pull-down approach (Dataset EV1). As expected, PTL does not bind α- nor β-tubulin in a specific manner when cells are treated with the compound before lysis (in cellulo) (Fig. 4A). However, in line with our in vitro microtubule polymerization assay (Appendix Fig. S1A) and previously published data (Hotta et al, 2021), tubulin becomes one of the targets when PTL is added directly to the cell lysate (in vitro) (Appendix Fig. S3D; Dataset EV2).

Interestingly, even though we used asynchronous cells for the pull-downs, mitotic proteins were notably represented (Appendix Fig. S3C), in agreement with the spindle and kinetochore binding of PTL and its effect on cell division. A total of 20 proteins with functions linked to mitosis were identified to be targeted by PTL (Table EV1). Among these proteins, only three are involved in kinetochore-microtubule attachment, namely CKAP5/chTOG, TPX2, and ZNF207/BUGZ (Fig. 4A).

CKAP5/chTOG is a microtubule and spindle assembly factor that localizes at kinetochores in mitosis (Herman et al, 2020). Although it regulates the dynamics of k-fibers (Barr and Gergely, 2008; Cassimeris et al, 2009; Herman et al, 2020), CKAP5 is also required for centrosomal microtubule nucleation in interphase (Ali et al, 2023) and its depletion induces multipolar spindles (Cassimeris and Morabito, 2004; Gergely et al, 2003), both of the functions that are not associated with PTL.

TPX2 is a spindle assembly factor that localizes at the mitotic spindle near the spindle poles (Garrett et al, 2002; Gruss et al, 2002; Wittmann et al, 2000). Even though it does not localize to

kinetochores, TPX2 contributes to the formation of k-fibers, mainly through its ability to enhance branching microtubule nucleation (Petry et al, 2013). However, similar to CKAP5, and in contrast to the PTL-induced pehenotype, TPX2 depletion results in the formation of multipolar spindles (Brunet et al, 2004; De Luca et al, 2006; Garrett et al, 2002; Goshima, 2011; Wittmann et al, 2000).

ZNF207/BUGZ is a zinc finger protein that localizes at the mitotic spindle and kinetochores. It is required for the loading and stabilization of BUB3 at kinetochores (Jiang et al, 2014; Toledo et al, 2014). In early mitosis, BUGZ binds to kinetochores, from which it is evicted in late prometaphase/metaphase (Shirnekhi et al, 2020). While it interacts with microtubules and promotes spindle assembly during mitosis, BUGZ localizes inside the nucleus in interphase. Due to its role in the formation of kinetochore-microtubule attachments, depletion of BUGZ causes severe problems in chromosome congression (Jiang et al, 2014; Toledo et al, 2014).

Based on this analysis of the localization and knockdown phenotypes of these candidate proteins, we focused on BUGZ, as the candidate that most closely resembles the cellular localization and phenotype of PTL.

## Parthenolide targets BUGZ by covalently binding to its microtubule-binding domain

To validate its cellular localization, we performed live-cell imaging of HeLa cells stably expressing GFP-BUGZ (Shirnekhi et al, 2020) (Appendix Fig. S3E). As previously reported (Shirnekhi et al, 2020), GFP-BUGZ displayed a clear kinetochore localization in early mitosis and was removed from kinetochores during cell's transition to metaphase (Fig. EV2A; Movie EV2). We next utilized RNAi in U2OS H2B-GFP/mScarlet-α-tubulin cells to validate the BUGZ knockdown phenotype. In agreement with earlier studies (Jiang et al, 2014; Toledo et al, 2014), and similar to the effect of 15 µM PTL, BUGZ depletion induced severe problems in chromosome congression that caused mitotic arrest (Fig. EV2B and Appendix S3F; Movie EV3). The mitotic arrest was caused by unattached chromosomes that failed to satisfy spindle assembly checkpoint, as revealed by live-cell imaging of HeLa cells stably expressing GFP-Mad2 (Appendix Fig. S4; Movie EV4). Moreover, tubulin intensity at the mitotic spindle after the cold treatment ($0.35 \pm 0.2$ in PTL and $0.52 \pm 0.26$ in siBUGZ, compared to $1.06 \pm 0.27$ in DMSO, and $1 \pm 0.29$ in siNT) (Fig. EV3A,B) and astrin intensity at congressed kinetochores ($0.7 \pm 0.25$ in PTL and $0.54 \pm 0.21$ in siBUGZ, compared to $1.04 \pm 0.26$ in DMSO and $1.01 \pm 0.19$ in siNT) (Fig. EV3C,D) were strongly reduced in BUGZ-depleted HeLa cells, resembling the effect of PTL on kinetochore-microtubule attachments. Thus, we confirmed that BUGZ is a kinetochore protein required for accurate kinetochore-microtubule attachment and chromosome congression in mitosis.

Next, we compared the effect of PTL in HeLa cells to its effect in HeLa GFP-BUGZ cells, which express at least 2-fold more BUGZ according to their immunoblot profile (Fig. 4B; Appendix S3E). Whereas 15 µM PTL induced severe chromosome congression problems followed by mitotic arrest in HeLa cells, a 3-fold higher concentration of PTL was required to induce similar effects in around 50% of BUGZ-overexpressing cells (Fig. 4C; Movies EV5 and EV6). Al.though this rescue of PTL phenotype by excess of

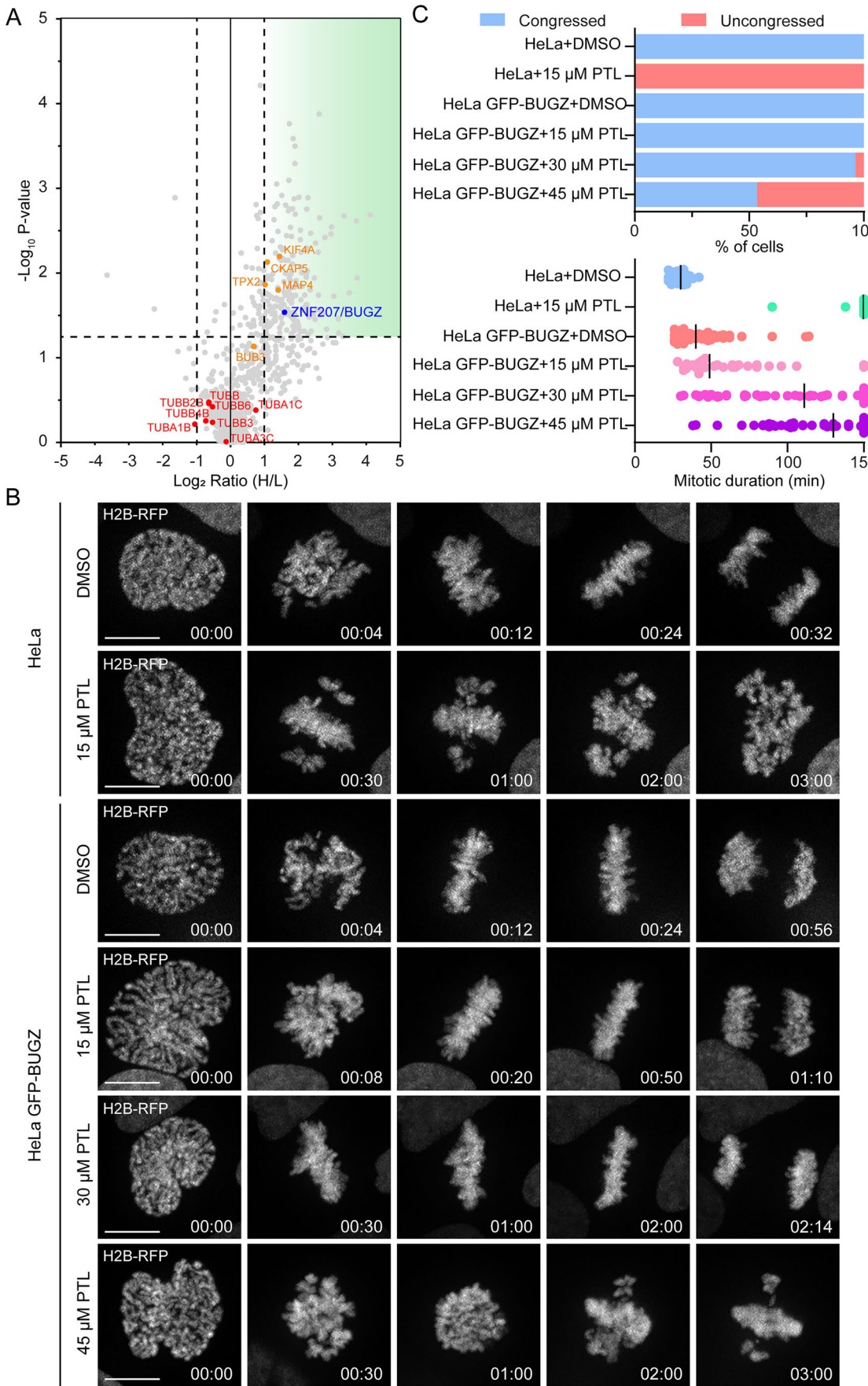

◄ **Figure 4. Parthenolide binds to kinetochore and spindle-associated proteins.**

(A) Volcano plot showing quantitative mass-spectrometry results. Dashed horizontal line shows the *p*-value cut-off (*p* < 0.05) and vertical dashed lines indicate the upregulated/downregulated (competed/non-competed by free PTL) proteins. The green transparent region groups all the proteins that satisfy the *p*-value cut-off and are upregulated (competed) with a SILAC ratio higher than 2. *N* (number of experiments): 3. Statistical analysis was performed using unpaired t-test. (B) Representative spinning disk confocal time-series of mitosis in HeLa parental cells and HeLa stably expressing GFP-BUGZ undergoing indicated treatments. Scale bar: 10 μm. (C) Quantification of chromosome congression status and mitotic duration in HeLa parental cells and HeLa stably expressing GFP-BUGZ undergoing indicated treatments. Median is plotted for mitotic duration. *N*, *n* (*N* = number of cells, *n* = number of experiments) for congression phenotype: HeLa + DMSO (44, 3), HeLa + 15 μM PTL (41, 3), HeLa GFP-BUGZ + DMSO (85, 3), HeLa GFP-BUGZ + 15 μM PTL (30, 3), HeLa GFP-BUGZ + 30 μM PTL (56, 3), HeLa GFP-BUGZ + 45 μM PTL (58, 3); *N*, *n* (*N* = number of cells, *n* = number of experiments) for mitotic duration: HeLa + DMSO (40, 3), HeLa + 15 μM PTL (36, 3), HeLa GFP-BUGZ + DMSO (85, 3), HeLa GFP-BUGZ + 15 μM PTL (30, 3), HeLa GFP-BUGZ + 30 μM PTL (56, 3), HeLa GFP-BUGZ + 45 μM PTL (58, 3). Source data are available online for this figure.

BUGZ suggests that BUGZ is a critical mitotic target of PTL, it is also possible that the overexpression of any PTL-binding protein can sequester PTL and thus rescue the phenotype by preventing its binding to other targets. We tested this by overexpressing three other PTL-binding mitotic proteins: CKAP5, TPX2, and KIF4A, and assessing their effect on PTL-induced mitotic problems. The overexpression of none of these three proteins was able to rescue the PTL-associated mitotic phenotypes, indicating that the rescue observed in HeLa GFP-BUGZ cells was indeed BUGZ-specific (Fig. EV4A–D; Movie EV7).

To further prove BUGZ as a PTL target, we immunoprecipitated FLAG-BUGZ from HEK 293T cells previously treated with alkyne-PTL and performed a click-chemistry reaction with fluor-azide for *in-gel* visualization (Figs. 5A and EV5A). Due to the covalent nature of the compound, targeted proteins become fluorescent after the click reaction, enabling their detection by SDS-PAGE. PTL covalently binds FLAG-BUGZ in cellulo, confirming the LC-MS data (Figs. 5A and EV5A).

To test whether BUGZ is indeed the main kinetochore target of PTL, we used fluorescence microscopy to monitor alkyne-PTL localization in control and BUGZ-depleted U2OS and HeLa GFP-BUGZ cells. Importantly, BUGZ and PTL fluorescence signals at single kinetochores clearly correlated, and alkyne-PTL localization at kinetochores was strongly reduced in BUGZ-depleted cells (by up to 75%), indicating that BUGZ is the main kinetochore target of PTL (U2OS: 1 ± 0.6 in siNT and 0.54 ± 0.4 in siBUGZ; HeLa GFP-BUGZ: 1.02 ± 0.27 in siNT and 0.27 ± 0.18 in siBUGZ) (Figs. 5B,C and EV5B–E). The remaining PTL signal at kinetochores may result from additional binding to other kinetochore target, such as CKAP5/chTOG identified in this study, and/or from insufficient RNAi-based depletion.

BUGZ is a highly disordered protein, containing a GLEBS domain, required for its interaction with BUB3 and consequent localization of BUB3, BUB1 and BUBR1 at kinetochores, and a microtubule binding domain in its N-terminal region (1–92) that is formed by two consecutive Cys2His2 zinc finger domains (Fig. EV5F). To test whether PTL interferes with the kinetochore scaffold function of BUGZ, we measured the kinetochore levels of BUB1 in nocodazole-arrested cells that were treated with DMSO as control or PTL. BUB1 intensity at kinetochores was not altered upon the PTL treatment, demonstrating that PTL does not affect the BUGZ-mediated recruitment of BUB1 and that its effect on mitosis is independent of the BUGZ role in the loading of BUB proteins to kinetochores (1.1 ± 0.4 in PTL, compared to 1.1 ± 0.28 in DMSO) (Fig. 5D,E).

In order to map the exact PTL-targeting residue of BUGZ, we performed LC-MS analysis of purified GST-BUGZ-compound

conjugates (Appendix Fig. S5A), as well as of PTL-bound GFP-BUGZ pulled-down using a GFP-Trap system in HeLa GFP-BUGZ cells (Fig. EV5G). Using both approaches, we detected a covalent PTL-BUGZ adduct formation within the peptide LYGP-GLAIHCMQVHK (Figs. 5F,G and EV5H). This peptide contains a Cys residue (Cys54) that belongs to the second zinc finger domain within the microtubule binding region of BUGZ. Cys54 is placed between His53 and His58 that are directly involved in zinc coordination (Fig. 5H). Given that PTL covalently targets a zinc finger domain within the BUGZ microtubule binding region, we propose that PTL may disrupt the structural properties of this domain, and particularly chelation of His53 to zinc due to the increased steric bulk of the adduct formed at neighboring Cys54, thereby preventing protein-protein interactions. Therefore, by binding to its microtubule-binding domain, PTL may interrupt the anchoring of BUGZ to microtubules and consequently obstruct kinetochore-microtubule attachment.

To provide structural insights into this disruption of protein-protein interactions we used AlphaFold 3 to generate a structure of BUGZ. This gave rise to a well-defined structure with a folded zinc finger with two zinc-binding domains, even in the absence of Zn²⁺. The structure was analyzed further in the Molecular Operating Environment (MOE, CCG). The two Zn²⁺ were inserted and the structure of Cys-bound PTL was retrieved from Protein Data Bank (6OCH) and aligned, with Cys54 providing a near perfect fit into a hydrophobic pocket in the zinc finger domain. The residues not in contact with PTL were fixed and the complex was relaxed by minimization and molecular dynamics calculations. The bound PTL could readily be replaced with the alkyne-PEG conjugate and the unbound PTL (Fig. 5H; Appendix S5B,C), with these all forming stable complexes. The complex of the unreacted PTL was subjected to long unconstrained molecular dynamics calculations at 300 K in saline and showed excellent stability. The resulting structures were analyzed with respect to important contacts and mechanism of reaction. These data are consistent with the hydrophobic PTL binding pocket being involved in the BUGZ interaction with microtubules thus allowing reaction with PTL to block this interaction as observed above.

To further investigate the role of Cys54 of BUGZ and to determine whether the Cys54 mutant phenocopies the PTL phenotype or fails to rescue it, we generated HeLa cell lines stably expressing GFP-tagged wild-type (WT) BUGZ and the C54A mutant (Appendix Fig. S6A,B). Using live-cell imaging, we observed that expression levels of the C54A mutant were consistently lower than those of BUGZ WT. Moreover, approximately 50% of cells expressing the RNAi-resistant C54A mutant

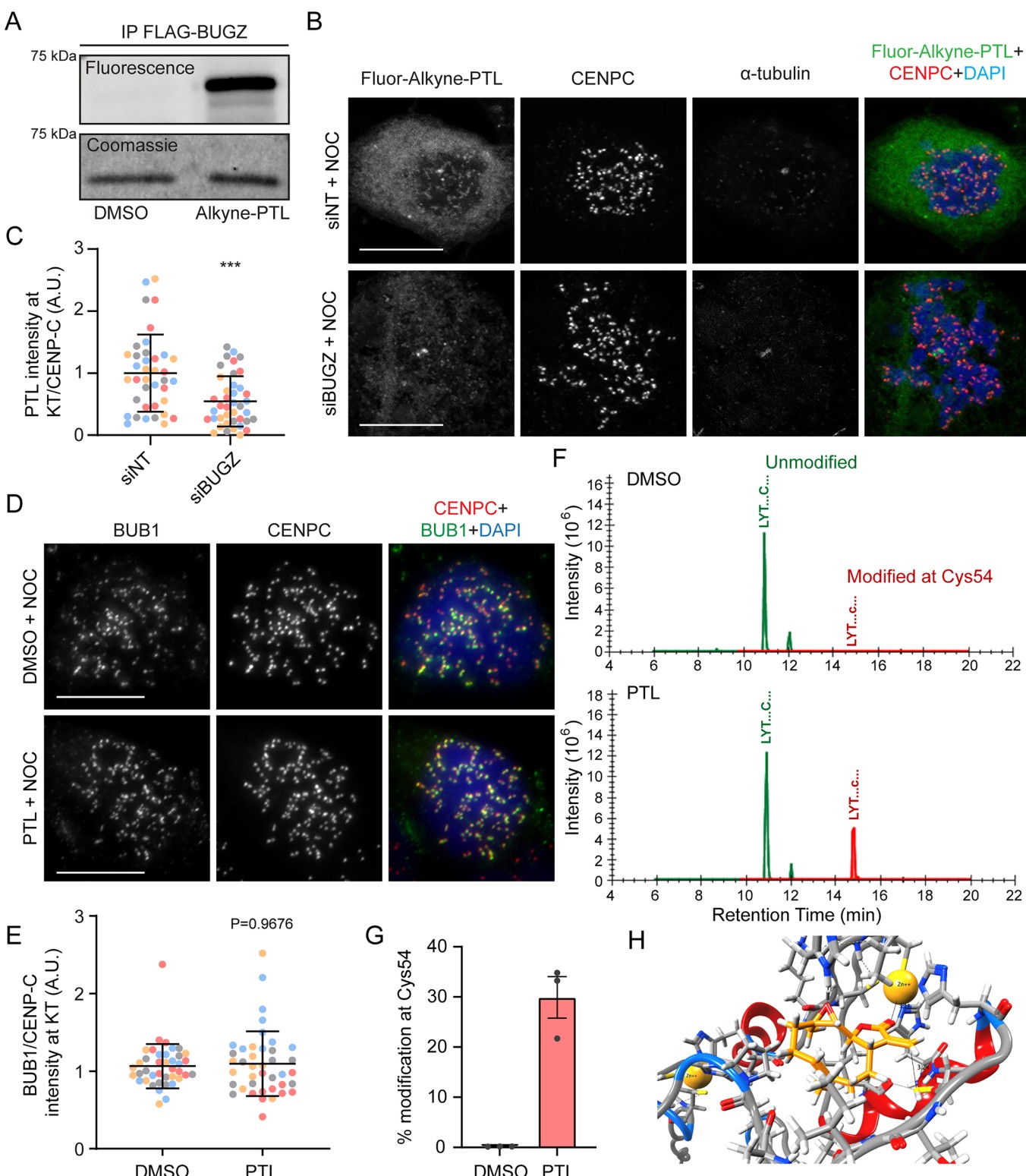

failed to rescue the depletion of endogenous BUGZ, exhibiting chromosome congression defects similar to those observed in BUGZ-depleted or PTL-treated cells (Appendix Fig. S7A,B). These findings suggest that the Cys54 mutation disrupts the zinc finger domain of BUGZ, impairing its mitotic function and preventing high cellular expression of the mutant protein. Consequently, we were unable to evaluate the ability of C54A mutant to bind PTL, since the mutation itself compromises BUGZ function.

◄ **Figure 5. Parthenolide targets BUGZ through covalent binding of the microtubule binding domain.**

(A) Fluorence and Coomassie staining of immunoprecipitated FLAG-BUGZ from HEK 293T cells treated either with DMSO or 15 µM alkyne-PTL. (B) Representative spinning-disk confocal maximum projections of click-based imaging of 15 µM fluor-488-alkyne-PTL in nocodazole arrested U2OS cells co-stained with α-tubulin and CENP-C, as markers for microtubules and kinetochores, respectively. Scale bar: 10 µm. (C) Quantification of the relative levels of 15 µM fluor-488-alkyne-PTL at kinetochores. N, n (number of cells, number of experiments): siNT (38, 4) siBUGZ (40, 4). ***p ≤0.001. (D) Representative confocal maximum projections of BUB1 immunostainings in nocodazole arrested U2OS cells treated with DMSO or 15 µM PTL. Scale bar: 10 µm. (E) Quantification of the relative levels of BUB1 at kinetochores. N, n (N = number of cells, n = number of experiments): DMSO (40, 4), 15 µM PTL (40, 4). (F) Extracted ion chromatograms for peptides with and without PTL modification at Cys54 (red, and green, respectively). (G) Extent of PTL-modification at Cys54 from 3 independent experiments. (H) AlphaFold 3 model of BUGZ Zinc Finger domains and Cys54 positioning showing the catalytic mechanism leading to PTL selectivity. Replicates are color-coded for all quantifications. Data in (C) and (E) are presented as mean ± SD values, while data in (G) are presented in mean ± SEM values. Statistical analysis was performed by unpaired t-test with Welch's correction in (C) and Mann-Whitney test in (E). Source data are available online for this figure.

## Discussion

PTL and its derivatives have shown highly promising anticancer effects in numerous preclinical studies. However, their usage in clinical trials has been limited, mainly due to their low bioavailability (Freund et al, 2020). Although total synthesis of PTL can be achieved (Freund et al, 2019), this has been challenging and large-scale synthesis beyond mg (10 mg) quantities have not yet been achieved. Moreover, the exact anticancer mode of action of PTL remains elusive. Thus, understanding the mechanism of interaction between PTL and its targets responsible for the anticancer activity is crucial for the development of more efficient PTL analogs with improved pharmacokinetic properties.

Here, we combined click chemistry, proteomics, and cell biology to analyze the long-overlooked antimitotic effect of PTL. We demonstrate that PTL binds kinetochores during early mitosis and interferes with the establishment of kinetochore-microtubule attachments required for accurate chromosome congression to the spindle equator and subsequent segregation of sister chromatids. Importantly, despite its ability to bind tubulin when used in in vitro assays ((Hotta et al, 2021) and this study), we show that PTL does not interfere with cellular microtubule dynamics and, therefore, does not act as a microtubule-targeting agent in cells, highlighting the importance of in cellulo validation of the in vitro-identified drugs targets. On the contrary, we identified kinetochore protein ZNF207/BUGZ as the main target of PTL in mitosis. We demonstrate that PTL binds to Cys54 of BUGZ within the second zinc finger domain of the BUGZ microtubule-targeting region. Comparing the antimitotic and anti-microtubule detyrosination activities of PTL and its analog costunolide, we propose that PTL covalently binds BUGZ via Michael addition of a Cys54 of BUGZ to the α-methylene-γ-lactone of PTL.

The molecular modeling presented in this work provides a mechanistic explanation for the high selectivity of PTL. The structures shown in Fig. 5H and Appendix S5B,C indicate a mechanism in which PTL is bound and coordinated to BUGZ by interactions directing and catalyzing the stereo-selective Michael addition of the Cys54 sulfhydryl to the carbonyl conjugated double bond. Thus, the coordination of the carbonyl group to $Zn^{2+}$, facilitated by the two hydrogen bonds between the epoxide and His38, activate the double bond for nucleophilic attack, thereby significantly increasing the selectivity of PTL. A detailed analysis of the interactions shows that PTL has close contact with His38, Leu50, His53, Thr61, Ile62, Ala64, Pro66, and Ile64, providing a hydrophobic environment for the binding to (hydrophobic) PTL. The distance between the carbonyl group and the $Zn^{2+}$ is 1.88, 3.20,

and 3.10 Å in the Cys54-bound PTL, the Cys54-bound PTL-PEG-conjugate, and unreacted PTL, respectively. The double bond in PTL is ~3.20 Å from the hydrogen-bonded sulfur nucleophile in the unreacted complex.

Altogether, we propose that PTL interferes with the ability of BUGZ to bind microtubules, consequently obstructing the establishment of proper kinetochore-microtubule attachment and leading to mitotic arrest.

In addition to BUGZ, we identified kinetochore-based CKAP5/chTOG and mitotic spindle-associated TPX2 as additional proteins to which PTL binds, and which could contribute to the mitotic effect of PTL and its interference with kinetochore-microtubule attachment. However, unlike PTL, knockdown of CKAP5 disrupts centrosomal microtubule nucleation in interphase (Ali et al, 2023), and depletion of either CKAP5 or TPX2 induces multipolar spindles (Brunet et al, 2004; Cassimeris and Morabito, 2004; De Luca et al, 2006; Garrett et al, 2002; Gergely et al, 2003; Goshima, 2011; Wittmann et al, 2000), another phenotype that is not characteristic of PTL activity. Importantly, we demonstrated that while BUGZ overexpression prevented PTL-induced chromosome congression defects, overexpression of CKAP5 or TPX2 failed to rescue the associated mitotic phenotypes. Thus, although we cannot completely rule out their partial contribution, based on their localization and detailed knockdown and overexpression phenotypes, we conclude that CKAP5 and TPX2 are unlikely to be the main targets responsible for the antimitotic effect of PTL. Noteworthy, PTL may bind to CKAP5 and TPX2 without interfering with their function, as well as it may interact with them indirectly, via one of their binding partners. The well-documented effect of PTL on microtubule detyrosination (Barisic et al, 2015; Fonrose et al, 2007), and consequently on kinetochore-based motor protein CENP-E (Barisic et al, 2015), may additionally explain its antimitotic activity. However, in our screen we did not identify an obvious target of PTL that could explain its ability to decrease microtubule detyrosination, and therefore this question remains to be answered by the future studies.

Given that BUGZ orthologues are conserved in eukaryotes, but not in lower organisms, such as yeasts, it is likely that BUGZ evolved to facilitate kinetochore functions in more complex systems (Toledo et al, 2014). Comparative functional genetic screening in patient-derived glioblastoma multiforme stem-like cells (GSC) and fetal neural stem cells revealed BUGZ as a candidate whose loss specifically prevented GSC proliferation, suggesting that glioblastoma cells have certain kinetochore signaling defects that can be overcome by BUGZ activity (Toledo et al, 2014). The finding that nontransformed cells are less sensitive to BUGZ loss/inhibition

compared to glioblastoma cells places BUGZ as a novel anticancer target for glioblastoma and other cancers with a compromised kinetochore-microtubule interface (Herman et al, 2015; Toledo et al, 2014). Thus, our identification of PTL as a BUGZ inhibitor may not only help elucidate the anticancer mechanism of this compound but may also provide important knowledge for future structural modifications of PTL and its derivatives, as well as future design of new BUGZ inhibitors as anticancer agents. Importantly, development of new antimitotic compounds with anticancer activity that do not target tubulin will avoid neurotoxicity and neuropathies associated with microtubule-targeting agents, caused by their interference with microtubule-dependent intraneuronal transport. In addition to its anti-inflammatory and ROS-associated effects, elucidating the antimitotic activity of PTL may help us to understand and improve the anticancer potential of PTL.

# Methods

### Reagents and tools table

| Reagent/Resource | Reference or Source | Identifier or Catalog Number |
|---|---|---|
| **Experimental models** | | |
| U2OS H2B-GFP | Barisic et al, 2010 | N/A |
| U2OS H2B-GFP/ mScarlet-α-tubulin | This study | N/A |
| U2OS EB1-GFP | Vinopal et al, 2012 | N/A |
| U2OS-PA-GFP/ mCherry-α-tubulin | van Heesbeen et al, 2014 | N/A |
| HeLa Kyoto | Danish Cancer Institute | N/A |
| HeLa GFP-BUGZ | Shirnekhi et al, 2020 | N/A |
| HeLa GFP-MAD2 | Schweizer et al, 2013 | N/A |
| HeLa hsCKAP5-LAP | Poser et al, 2008 | N/A |
| HeLa hsTPX2-GFP | Poser et al, 2008 | N/A |
| HeLa GFP-BUGZ | This study | N/A |
| Rosetta2 | MilliporeSigma | 71402 |
| **Recombinant DNA** | | |
| pFA6a-mScarlet-mNeonGreen-kanMX | Addgene; A. Khmelinskii | 173454 |
| pENTR4 no ccDB | Addgene; E. Campeau | 17424 |
| pENTR4-GFP | Addgene | 17396 |
| pENTR4-GFP-BUGZ | This study | N/A |
| pENTR4-GFP-BUGZ C54A | This study | N/A |
| pDONR207 TubA1B | This study | N/A |
| pGenDONR-BUGZ | This study | N/A |
| pGenDONR-BUGZ C54A | This study | N/A |
| pENTR4-mScarlet-I | This study | N/A |
| pENTR4-mScarlet-I-TubA1B | This study | N/A |
| pLenti CMV Puro DEST | Addgene; E. Campeau | 17452 |

| Reagent/Resource | Reference or Source | Identifier or Catalog Number |
|---|---|---|
| pLenti CMV Puro mScarlet-I-TubA1B | This study | N/A |
| pLenti CMV/TO puro DEST | Addgene | 17293 |
| pLenti CMV/TO puro GFP-BUGZ | This study | N/A |
| pLenti CMV/TO puro GFP-BUGZ C54A | This study | N/A |
| GUT-BUGZ-FL | Addgene | 84024 |
| pDNOR-207 | Stephan Geley, Medical University of Innsbruck, Austria | N/A |
| pGEX-BUGZ | This study | N/A |
| ΔDest-FLAG-BUGZ | This study | N/A |
| **Antibodies** | | |
| Human anti-CREST | Stephan Geley, Medical University of Innsbruck, Austria | N/A |
| Guinea pig anti-CENP-C | MBL | PD030 |
| Rabbit anti-astrin | Bethyl Lab | A301-511A |
| Mouse anti-α-tubulin | Sigma-Aldrich | T5168 |
| Rabbit anti-centrin | Iain Cheeseman, Whitehead Institute for Biomedical Research, Cambridge, MA, USA | N/A |
| Rabbit anti-CENP-E | Abcam | Ab133583 |
| Mouse anti-BUB1 | Abcam | ab181438 |
| Rabbit anti-BUGZ | Sigma-Aldrich | HPA017013 |
| Mouse anti-GAPDH | Proteintech | 60004 |
| Mouse anti-vinculin | Sigma-Aldrich | SAB4200729 |
| Rabbit anti-detyrosinated tubulin | Liao et al, 2019 | N/A |
| Mouse anti-FLAG | Sigma-Aldrich | F3165 |
| ChromoTek GFP-booster | Proteintech | gba488-100 |
| Goat anti-Mouse IgG (H+L) Alexa Fluor™ 488 | Life Technologies | A11029 |
| Goat anti-Rabbit IgG (H+L) Alexa Fluor™ 488 | Life Technologies | A11034 |
| Goat anti-Mouse IgG (H+L) Alexa Fluor™ 568 | Life Technologies | A11031 |
| Goat anti-Rabbit IgG (H+L) Alexa Fluor™ 568 | Life Technologies | A11036 |
| Goat anti-Human IgG (H+L) Alexa Fluor™ 568 | Life Technologies | A21090 |
| Goat anti-Guinea pig IgG (H+L) Alexa Fluor™ 568 | Life Technologies | A11075 |

| Reagent/Resource | Reference or Source | Identifier or Catalog Number |
|---|---|---|
| Goat anti-Guinea pig IgG (H+L) Alexa Fluor™ 647 | Life Technologies | A21450 |
| Goat anti-mouse HRP | Jackson ImmunoResearch | 115-035-003 |
| Goat anti-rabbit HRP | Jackson ImmunoResearch | 111-035-003 |
| Rabbit anti KIF4A | Bethyl Lab | A301-074A |
| Rabbit anti CKAP5 | Abcam | ab86073 |
| **Oligonucleotides and other sequence-based reagents** | | |
| PCR primers | This study | N/A |
| siNT | Lavrsen et al, 2023 | N/A |
| siBUGZ | Jiang et al, 2014 | N/A |
| **Chemicals, Enzymes and other reagents** | | |
| DMEM | Thermo Fisher | 31966-047 |
| Opti-MEM | Thermo Fisher | 31985-070 |
| Fetal bovine serum (FBS) | Thermo Fisher | A5256701 |
| DMSO | Sigma-Aldrich | 5895690100 |
| Lipofectamine RNAiMAX | Thermo Fisher | 13778150 |
| Metafectene | Biontex | T020-1.0 |
| GatewayTM LR ClonaseTM II Enzyme mix | Invitrogen | 11791020 |
| pAd/CMV/V5-DEST Gateway Vector Kit | Thermo Fisher | V49320 |
| Parthenolide | Santa Cruz Biotechnology | sc-3523 |
| Alkyne-Parthenolide | This study | N/A |
| Costunolide | Sigma-Aldrich | SML0417 |
| Nocodazole | Sigma-Aldrich | M1404 |
| Apcin | Sigma-Aldrich | SML1503 |
| proTAME | R&D Systems | I-440-01M |
| MG132 | Sigma-Aldrich | 474790 |
| Ampicilin | Sigma-Aldrich | A8351 |
| Chloramphenicol | Sigma-Aldrich | C0378 |
| IPTG | Millipore | 420322 |
| EDTA free complete protease inhibitor | Sigma-Aldrich | 4693132001 |
| Lysozyme | Sigma-Aldrich | 62970 |
| DNAseI | Sigma-Aldrich | 000000010104159001 |
| Methanol | Merck | MERC1.06009 |
| Paraformaldehyde (PFA) | Electron Microscopy Sciences | 15710 |
| SiR-DNA | Spirochrome | SC007 |
| Phalloidin-iFluor 555 | Abcam | Ab176756 |
| DAPI | Sigma-Aldrich | D9542 |
| mCherry-KIF4A adenovirus | Steblyanko et al, 2020 | N/A |
| H2B-RFP adenovirus | Eibes et al, 2023 | N/A |

| Reagent/Resource | Reference or Source | Identifier or Catalog Number |
|---|---|---|
| H2B-GFP adenovirus | This study | N/A |
| Click-iT Plus Alexa Fluor Picolyl Azide Toolkit | Thermo Fisher | C10642 |
| Fluor-488 azide | Sigma-Aldrich | 760765 |
| Fluor-cy5 azide | Sigma-Aldrich | 777323 |
| Heavy isotope-labeled arginine ($^{13}C_6$, $^{15}N_4$-arginine) | Sigma-Aldrich | 608033 |
| Heavy isotope-labeled lysine ($^{13}C_6$, $^{15}N_2$-lysine) | Cambridge Isotope Laboratories | CNLM-291-H-0.5 |
| Dialyzed serum | Sigma-Aldrich | F0392 |
| Glass bottom 35 mm dishes | MatTek | P35G-1.5-14-C |
| Round 12 mm cover glasses | Menzel Glaser | MENZCB00120RAC20 |
| Azide-sepharose beads | Kerafast | FCC422 |
| Dynabeads Protein G | Thermo Fisher | 10003D |
| ChromoTek GFP-Trap agarose beads | Proteintech | gta-20 |
| BCA assay kit | Pierce | 23225 |
| ECL Western Blotting Substrates | Bio-Rad | 1705060 |
| Sep-pak C18 cartridge | Waters | WAT054955 |
| GST Sepharose 4 fast flow column | GE Healthcare | GE17-5132-01 |
| Superdex S200 increase 10/300 GL column | GE Healthcare | GE28-9909-44 |
| **Software** | | |
| ImageJ | https://imagej.net/software/fiji | N/A |
| GraphPad Prism 8.0 | https://www.graphpad.com | N/A |
| MaxQuant | https://www.maxquant.org | N/A |
| MSFragger (v3.7) | https://msfragger.nesvilab.org | N/A |
| FragPipe (v19.1) | https://fragpipe.nesvilab.org | N/A |
| Skyline (v23.1.0.268) | https://skyline.ms | N/A |
| Adobe Illustrator | https://www.adobe.com/ | N/A |
| BioRender | https://www.biorender.com/ | N/A |
| **Other** | | |
| Spinning-disk confocal microscope | 3i | N/A |
| LSM700 | Zeiss | N/A |
| LSM800 | Zeiss | N/A |

| Reagent/Resource | Reference or Source | Identifier or Catalog Number |
|---|---|---|
| Proxeon easy nLC system with Q-Exactive HF mass spectrometer | Thermo Fisher | N/A |
| Bruker timsTOF Pro mass spectrometer | Bruker Daltonics | N/A |
| Bruker Impact-II in positive ion mode with ESI | Bruker Daltonics | N/A |
| Dionex Ultimate 3000 | Thermo Fisher | N/A |

## Cell culture, transfection, treatments, and cloning

Cells were cultured in Dulbecco's modified Eagle medium (DMEM, Thermo Fisher) supplemented with 10% Fetal Bovine Serum (FBS, Thermo Fisher) in a humidified $CO_2$ (5%) incubator at 37 °C. Cells were regularly tested for mycoplasma contamination.

U2OS cell line stably expressing H2B-GFP/mScarlet-α-tubulin was created as follows. Firstly, mScarlet-I was amplified by PCR from pFA6a-mScarlet-mNeonGreen-kanMX (gift from Anton Khmelinskii, Addgene plasmid #173454) using the primers 5′-ATGGGAACCAATTCGCCACCATGGTGAGCAAGGGCGAG-3′ and 5′-CGCGGATCCCTTGTACAGCTCGTCCAT-3′ and cloned into pENTR4 no ccDB plasmid (gift from E. Campeau, Addgene plasmid #17424). Tubulin A1B was subcloned from pDONR207 TubA1B and cloned into pENTR4-mScarlet-I using KpnI and XhoI sites to create pENTR4-mScarlet-I-TubA1B vector. Subsequently, mScarlet-I-TubA1B was subcloned into pLenti CMV Puro DEST (gift from E. Campeau, Addgene plasmid #17452) by LR recombination (Invitrogen) according to manufacturer's instructions. Lentivirus expressing mScarlet-I-TubA1B was used to infect U2OS H2B-GFP cells (Barisic et al, 2010) (kind gift from Stephan Geley, Medical University of Innsbruck, Austria) and following antibiotic treatment, single-cell clonal selection was performed to obtain U2OS H2B-GFP/mScarlet-α-tubulin cells.

HeLa cells stably expressing GFP-BUGZ and GFP-BUGZ C54A were generated as follows. The BUGZ gene (human, NCBI RefSeq: NM_001032293.3) encoding wild-type BUGZ and C54A mutant were commercially synthesized (GenScript) as siRNA-resistant sequences into a pGenDONR vector (pGenDONR-BUGZ and pGenDONR-BUGZ C54A). To confer resistance to siRNA targeting the sequence 5′-GCCTGCTACACTTACAACAACTAGT-3′, silent mutations were introduced at codons corresponding to residues 336–343, replacing the original nucleotide sequence with 5′-GCCCGCAACCTTGACCACCACCTCC-3′ without altering the encoded amino acids. BUGZ variants were then amplified by PCR using the primers 5′-GGTGGTGGTACCATGGGTCGCAAGAAGAAGAAG-3′ and 5′-GGTGGTGGATCCTCAGTAACGGCCACCTTG-3′ to introduce KpnI and BamHI sites and clone them into pENTR4-GFP (Addgene # 17396) vector for Gateway-compatible N-terminal GFP tagging to generate pENTR4-GFP-BUGZ and pENTR4-GFP-BUGZ C54A plasmids. The GFP-BUGZ construct was later subcloned into pLenti CMV/TO Puro DEST (Addgene #17293) destination vector by LR recombination according to manufacturer's instructions, obtaining pLenti CMV/TO puro GFP-BUGZ plasmid. Lentiviruses containing GFP-BUGZ and GFP-BUGZ C54A were used to infect HeLa Kyoto cells. Following infection, cells were selected with puromycin, and serial dilutions were performed for clonal selection.

For siRNA-mediated depletion experiments, cells were transfected in OptiMEM (Thermo Fisher) with Lipofectamine RNAi-MAX (Thermo Fisher) with 50 nM siRNAs for 72 h – siBUGZ: 5′-GCCUGCUACACUUACAACAACUAGU-3′ (Jiang et al, 2014), and siNT (non-targeting control siRNA): 5′-UGGUUUACAUGUCGACUAA-3′.

For the analysis of efficiency of PTL and its derivatives in tubulin detyrosination inhibition, cells were treated with 3.3 μM nocodazole for 3 h to promote microtubule depolymerization. Later, microtubule repolymerization was permitted by thoroughly washing out nocodazole from the cells and the addition of fresh warm DMEM containing the indicated drugs (15 μM each) for a further 2 h.

For Gateway cloning, BUGZ entry vector was generated by amplification of BUGZ DNA sequence from GUT-BUGZ-FL (addgene #84024) using the following primers: BUGZ_BP_Fwd 5′-AAAAAAGCAGGCTCCATGGGTCGCAAG-3′ and BUGZ_BP_Rev 5′-CAAGAAAGCTGGGTTTCAGTAACGGCC-3′ and recombination into a pDNOR-207 plasmid (gift from Stephan Geley, Medical University of Innsbruck). BUGZ was later subcloned into a FLAG-DEST and pGEX-DEST (gift from Stephan Geley, Medical University of Innsbruck) vectors by LR reaction.

## Live cell imaging

Time-lapse imaging was performed in a heated incubation chamber at 37 °C with controlled humidity and 5% $CO_2$ supply using a Plan-Apochromat 63×/1.4 NA oil objective with differential interference contrast mounted on an inverted Zeiss Axio Observer Z1 microscope (Marianas Imaging Workstation (3i—Intelligent Imaging Innovations, Inc.)), equipped with a CSU-X1 spinning disk confocal head (Yokogawa Corporation of America) and four laser lines (405, 488, 561, and 640 nm). Images were acquired using an iXon Ultra 888 EM-CCD camera (Andor Technology).

To study the mitotic effect of PTL, HeLa Kyoto, HeLa GFP-BUGZ (Shirnekhi et al, 2020) (gift from Jeniffer DeLuca, Colorado State University, USA), HeLa GFP-MAD2 (Schweizer et al, 2013) (gift from Helder Maiato, i3S Porto, Portugal), U2OS H2B-GFP/mScarlet-α-tubulin, HeLa hsCKAP5-LAP and HeLa hsTPX2-GFP (Poser et al, 2008) (gift from Anthony Hyman, Max Planck Institute of Cell Biology and Genetics, Germany), or HeLa cells infected with adenovirus to express mCherry-KIF4A (Steblyanko et al, 2020) were seeded in glass bottom 35 mm dishes (MatTek) and DMSO (Sigma-Aldrich) or PTL (15, 30, or 45 μM) (Santa Cruz Biotechnology) were added 10 min before imaging. Adenovirus encoding H2B-GFP (AV-H2B-GFP) and H2B-RFP (AV-H2B-RFP) (Eibes et al, 2023) were produced using pAd/CMV/V5-DEST Gateway Vector Kit (Thermo Fisher) according to the manufacturer's instructions. For live-cell imaging, cells were infected with AV-H2B-RFP or AV-H2B-GFP over night before drug treatment to enable chromosome visualization. Cells were imaged every 2 min with 1 μm separation interval between z-planes, covering the entire mitotic spindle. All images represent the maximum-projections of z-stacks. Mitotic duration was quantified by tracking the time spent from nuclear envelope breakdown to anaphase onset, with

exception for the cells that remained in mitosis at the end of imaging, whose mitotic duration was scored as the time spent from nuclear envelope breakdown to the end of imaging.

To observe microtubule dynamics during interphase, U2OS EB1-GFP cells (Vinopal et al, 2012) (gift from Pavel Draber, Institute of Molecular Genetics of the Czech Academy of Sciences-IMG ASCR, Prague, Czech Republic) were treated with indicated drugs 15 min before imaging. To study microtubule dynamics during mitosis, cells were arrested in metaphase using anaphase-promoting complex/cyclosome (APC/C) inhibitors Apcin (20 μM, Sigma-Aldrich) and a cell permeable tosyl-L-arginine methyl ester (proTAME) (10 μM, R&D Systems) for 2 h before the addition of 15 μM PTL or DMSO vehicle control. DNA was labeled by adding 20 nM SiR-DNA (Spirochrome) 1 h prior to live-cell imaging. Cells were imaged at 2 frames per second rate for 1–2 min. 2–5 EB1 comets per cell were tracked manually using ImageJ.

Microtubule flux and turnover rates in mitotic spindles were imaged and quantified using U2OS-PA-GFP/mCherry-α-tubulin cells (van Heesbeen et al, 2014) (gift from Rene Medema, Netherlands Cancer Institute-NKI, Amsterdam, Netherlands) as previously described (Eibes et al, 2023). Cells were arrested in metaphase using 5 μM MG132 for 1 h followed by treatment with 15 μM PTL or DMSO vehicle control for 15 min.

Images were processed using ImageJ and Photoshop 2020 (Adobe).

## Immunofluorescence and click chemistry-based imaging

Cells were grown on 12 mm round cover glasses (Menzel Glaser) and fixed either by ice-cold methanol at −20 °C or pre-warmed 4% PFA in PHEM buffer (120 mM PIPES pH 6.9, 50 mM HEPES, 20 mM EGTA, 8 mM MgSO$_4$). The following primary antibodies were used in this study: human CREST antiserum 1:5000 (gift from Stephan Geley, Medical University of Innsbruck), guinea-pig anti-CENP-C 1:2000 (MBL), rabbit anti-astrin/MAP126 1:500 (Bethyl Lab), mouse-anti α-tubulin 1:2000 (Sigma-Aldrich), rabbit anti-centrin 1:2000 (gift from Iain Cheeseman, Whitehead Institute for Biomedical Research, Cambridge, MA, USA), rabbit CENP-E 1:500 (Abcam), mouse anti-BUB1 1:500 (Abcam), Alexa fluor 488, 568, 647 secondary antibodies (Invitrogen) were used at 1:1000. For DNA counterstaining DAPI was used at 1 μg/ml (Sigma-Aldrich).

For alkyne-PTL cell imaging, cells grown on coverslips were treated with 5 μM or 15 μM alkyne-PTL, pre-extracted with PHEM + 0.5% Triton-X buffer for 30 s and fixed by prewarmed 4% PFA in PHEM buffer (120 mM PIPES pH 6.9, 50 mM HEPES, 20 mM EGTA, 8 mM MgSO$_4$) for 10 min. Cells were blocked with immunostaining solution (0.5% Triton-X, 5% FBS in PBS) for 30 min. Click reaction was performed as follows: 84 μM THPTA (Sigma-Aldrich), 200 μM CuSO$_4$ (Sigma-Aldrich), 3 mM sodium ascorbate (Sigma-Aldrich), 5 μM Fluor-488 azide (Sigma-Aldrich) in PBS for 1 h at RT or by using Click-iT Plus Alexa Fluor Picolyl Azide Toolkit (Thermo Fisher) for 12.5 μM Fluor Cy5 azide (Sigma-Aldrich) in DMSO, following the manufacturer's protocol. As phalloidin binding is disrupted at a high copper concentration, for actin co-staining, THPTA, CuSO$_4$, and sodium ascorbate were reduced to 42 μM, 100 μM, and 1.5 mM, respectively. Once the click reaction was completed, coverslips were washed 3 times with PBS to continue with the immunostainings. Phalloidin-iFluor 555

(Abcam) was used at 1:1000 for 1 h and ChromoTek GFP-booster at 1:400 (Proteintech) to enhance the GFP signal.

## Image acquisition and analysis

For microtubule regrowth assays, cells were first pre-treated with either DMSO or 15 μM PTL for 2 h. After pre-treatment, microtubules were depolymerized by incubating U2OS cells on ice in presence of DMSO or PTL for 30 min. To induce microtubule regrowth, cells were incubated with warm DMEM media containing the specified drugs. Cells were immediately fixed with ice-cold methanol at indicated time points. Images were acquired using Zeiss AxioObserver Z1 wide-field microscope (63x Plan-Apochromatic oil differential interference contrast objective lens, 1.4 NA) equipped with Metal halide arc lamp and Axiocam 702 mono CMOS camera and Zen 3.0 blue edition software (Carl Zeiss, Inc., Oberkochen, Germany). For quantification of microtubule nucleation, α-tubulin intensity at 1 min and 5 min time-point was selected by a circular ROI covering the centrosome, followed by subtraction of the cytoplasmic background. The resulting intensity values were normalized to the median of control. Centrin was used to define the centrosomal position.

For quantification of BUB1 intensity at kinetochores, microtubules were depolymerized with 3.3 μM nocodazole for 3 h. Images were acquired using a Plan-Apochromat 63x/1.4 NA oil objective with differential interference contrast mounted on an inverted Zeiss Axio Observer Z1 microscope (Marianas Imaging Workstation, 3i-Intelligent Imaging Innovations, Inc., Denver, CO, USA) equipped with an ORCA-Flash4.0 Digital CMOS camera (Hamamatsu).

For quantification of microtubule populations in mitotic cells, U2OS parental cells were arrested in metaphase using APC/C inhibitors Apcin and proTAME as described above, followed by treatment with 15 μM PTL, 15 μM alkyne-PTL, 15 μM costunolide (Sigma-Aldrich) or DMSO for 1 h. Following the treatment, cells were fixed with 4% paraformaldehyde in PHEM buffer for 20 min at 37 °C and microtubules were stained using anti-α-tubulin antibody. Images were acquired using a Plan-Apochromat 63x/1.4 NA oil objective with differential interference contrast mounted on an inverted Zeiss Axio Observer Z1 microscope (Marianas Imaging Workstation, 3i-Intelligent Imaging Innovations, Inc., Denver, CO, USA) equipped with an iXon Ultra 888 EM-CCD camera (Andor Technology, Belfast, UK). Astral and spindle microtubule intensities were quantified using ImageJ (National Institute of Health, Bethesda, MD, USA) as previously described (Rajendraprasad et al, 2021). Metaphase spindle length was measured in ImageJ by quantifying the three-dimensional distance between the two spindle poles, using centrin signal as reference.

For quantification of astrin and alkyne PTL intensity at kinetochores, images were acquired as described in the previous paragraph. Intensities from at least 20 kinetochores per cell were measured. Fluorescence signal of the examined protein at each kinetochore was normalized to CENP-C or CREST fluorescence. The protein of interest/CENP-C intensity ratios were subsequently normalized to control mean value.

To quantify k-fiber intensity, DMSO or 15 μM PTL pre-treated U2OS or HeLa cells were incubated on ice for 5 min. Cells were immediately fixed with ice-cold methanol. K-fiber intensity was measured by drawing a ROI around the mitotic spindle and background subtraction from an adjacent cytoplasmic area.

Representative images were acquired using LSM700 or LSM800 confocal microscope (Carl Zeiss Inc., Oberkochen, Germany) mounted on a Zeiss-Axio imager Z1 equipped with plan-apochromat 63x/1.4NA oil DIC M27 objective (Carl Zeiss, Inc., Oberkochen, Germany) and Zen 2008 software (Carl Zeiss, Inc., Oberkochen, Germany).

## Alkyne PTL synthesis

See Appendix Supplementary Methods for detailed description.

## Pull down and fluorescence tagging

For SILAC labeling of U2OS cells, DMEM media was supplemented with regular arginine and lysine, or with heavy isotope-labeled arginine ($^{13}C_6$, $^{15}N_4$-arginine, Sigma-Aldrich) and lysine ($^{13}C_6$, $^{15}N_2$-lysine, Cambridge Isotope Laboratories) and dialyzed serum (Sigma). To ensure metabolic labeling, cells were maintained in the specified media for 10 passages. A total of $2 \times 150$ mm plates per condition were used for in vitro pull down, and $4 \times 150$ mm plates per condition for in cellulo pull down.

For in cellulo experiments, SILAC-light cells were pre-incubated with 60 μM PTL for one hour followed by 15 μM alkyne-PTL incubation of light and heavy cells for another hour. Cells were lysed with modified NP-40 buffer (1% NP-40, 150 mM NaCl, 50 mM Tris pH 8.0) and 3 freezing/thawing cycles with liquid $N_2$. For in vitro experiments, ~6 mg of light SILAC protein lysate were pre-incubated with 120 μM PTL at 4 °C for 1 h followed by 15 μM alkyne-PTL incubation of light and heavy SILAC lysates at 4 °C for 1 h. Following PTL and alkyne-PTL incubation, SDS at 0.1% was incorporated into the lysates to promote protein denaturation and enable accessibility of the beads. The click reaction with agarose beads was performed overnight at 4 °C using 840 μM THPTA, 2 mM CuSO$_4$, 30 mM sodium ascorbate and 80 μl of agarose azide beads (Kerafast). Beads were washed 3 times with PBS + 0.5% SDS.

For Fluor-488 labeling and gel analysis, click reaction was performed using 20 μl of protein lysate and 840 μM THPTA, 1 mM CuSO$_4$, 15 mM sodium ascorbate and 50 μM of Fluor-488 azide.

## Sample preparation and mass spectrometry

Azide-Sepharose beads were dissolved in 8 M guanidine hydrochloride (GuHCl), 50 mM HEPES, pH 8.0 and protein content measured by BCA assay (Pierce). Protein was reduced and alkylated by 5 mM Tris(2-carboxyethyl) phosphine hydrochloride (TCEP) and 5 mM chloroacetamide for 1 h at RT. Before addition of proteases the protein was diluted to 2 M GuHCl by addition of 50 mM HEPES, pH 8.0. Lys-C was added (1/200 w:w) and the protein digested for 2–4 h at 37 °C, the protein was further diluted to 1 M GuHCl, before digesting with trypsin protease (1/200 w:w) overnight at 37 °C. Peptides were purified using a Sep-pak C18 cartridge (Waters) and were eluted in 50% acetonitrile.

Peptide fractions were analyzed by online nanoflow liquid chromatography-coupled tandem mass spectrometry (nLC-MS/MS) using a Proxeon easy nLC system connected to a Q-Exactive HF mass spectrometer (Thermo Scientific).

The raw MS data was computationally processed using MaxQuant (developer version 1.5.5.4) and searched against the UniProt database (downloaded July 6, 2015) using the integrated Andromeda search engine (Cox et al, 2011). Protein quantification required a minimum of 2 peptides, 1 unique peptide, and 2 SILAC ratio counts.

## Protein purification

For GST-BUGZ purification, chemically competent Rosetta2 bacteria were transformed with pGEX-BUGZ plasmid by heat shock method. Overnight grown pre-culture was resuspended in 1 l of LB-amp-chloro and incubated at 37 °C at 150 rpm. Protein expression was induced by the addition of 1 mM IPTG and incubated at 16 °C overnight. Bacteria expressing the protein were harvested by centrifuging the cells at 1500 rcf at 4 °C for 10 min and frozen at −20 °C until use. The cell pellet was resuspended with 50 ml of the lysis buffer (Binding Buffer (50 mM HEPES pH 7.5, 300 mM NaCl, 10% Glycerol), 1 mM PMSF, EDTA free complete protease inhibitor (Sigma-Aldrich), 1 mg/ml of lysozyme (Sigma-Aldrich) and 100 mg/ml of DNAseI (Sigma-Aldrich) and sonicated for 5 min (Amp 40%). The lysate was centrifuged at 20,000 rcf for 30 min at 4 °C. The filtered supernatant was loaded onto a GST Sepharose 4 fast flow column (GE Healthcare) pre-equilibrated with BB. After 10CV wash, GST tagged protein was eluted from the column using 20 mM glutathione. The eluted protein was concentrated using MWCO 30000. The protein was loaded onto a pre-equilibrated Superdex S200 increase 10/300 GL column (Cytiva Life Sciences). The fractions were analyzed by SDS-PAGE.

For mass spectrometry analysis, 10 μM of GST-BUGZ was incubated with 100 μM PTL for 20 min at RT. For fluorescence tagging, the purified protein was incubated with alkyne-PTL for 20 min at RT. Click reaction was performed as follows: 84 μM THPTA, 200 μM CuSO$_4$, 3 mM sodium ascorbate and 5 μM Fluor-488 azide for 1 h at RT.

## Identification of modified BUGZ peptides by mass spectrometry

5 μg of DMSO- or PTL-treated BUGZ was subjected to SP3 bead-based cleanup for removal of excess PTL prior to reduction/alkylation (TCEP/CAA) and digestion overnight with trypsin. Cleanup was performed as described by Batth et al (Batth et al, 2019), with slight modifications; samples (50 μl) were precipitated onto magnetic beads by addition of acetonitrile (ACN, 120 μl), washed (800 μl, $1 \times 70\%$ EtOH, $2 \times 100\%$ ACN), resuspended in TEAB (50 μl). Protein was reduced and alkylated for 10 min at 70 °C with TCEP (10 mM) and CAA (40 mM), followed by overnight digestion with trypsin (0.1 μg) at 37 °C. An additional aliquot of trypsin was added, followed by incubation for 1 h at 37 °C. Elutions from PTL-treated GFP-BUGZ expressing cells were prepared as described above, without reduction/alkylation. Digested tryptic peptides were subjected to stage-tip solid-phase extraction on C18 discs and analyzed on a Bruker timsTOF Pro mass spectrometer (Bruker Daltonics) or Bruker Impact-II in positive ion mode with ESI (CaptiveSpray, or Apollo source, respectively) connected on-line to a Dionex Ultimate 3000 chromatography system (Thermo Fisher Scientific). The mass spectrometers were operated in data-dependent acquisition (DDA) mode. The fragmentation spectra were searched against a FASTA database of GST-tagged BUGZ plus common contaminants, or a FASTA database of the human proteome plus GFP-BUGZ and

common contaminants, using MSFragger (v3.7) implemented in FragPipe (v19.1). Default settings (peptide length 7–50) were used in the FragPipe data analysis workflow, with variable modifications for cysteine alkylation (+57), n-terminal acetylation (+42), PTL modification at cysteine (+248.1) and oxidation at methionine (+16). PTL-modified spectral matches were manually validated, with peak areas and extracted ion chromatogram prepared in Skyline (v23.1.0.268). MSMS spectral annotation was performed in Fragpipe-PDV, Skyline or xiSPEC. The data have been deposited to the ProteomeXchange Consortium via the PRIDE (Perez-Riverol et al, 2025) partner repository with the dataset identifier PXD063178.

## Immunoblotting and immunoprecipitation

For immunoblotting, cells following subjected treatments were collected and lysed in NP40 buffer (50 mM Tris–HCl pH 8.0, 150 mM NaCl, 5 mM EDTA, 0.5% NP-40, 1 × EDTA-free protease inhibitor (Sigma-Aldrich), 1 × phosphatase inhibitor cocktail (Sigma-Aldrich), 1 mM PMSF). Immunoblotting was performed as previously described (Steblyanko et al, 2020) and the membranes were incubated with the following primary antibodies: rabbit anti-BUGZ 1:1000 (Sigma-Aldrich), mouse anti-GAPDH 1:25,000 (Proteintech), mouse anti-vinculin 1:5000 (Sigma-Aldrich), rabbit anti-detyrosinated tubulin 1:10,000 (Liao et al, 2019), mouse anti-α-tubulin 1:2000 (Sigma-Aldrich), rabbit anti-CKAP5 1:2000 (Abcam), rabbit anti-KIF4A 1:2000 (Bethyl Lab). HRP-conjugated secondary antibodies were used at 1:10,000 (Jackson ImmunoResearch) and visualized by ECL (Bio-Rad).

To immunoprecipitate FLAG-BUGZ, HEK 293T cells were transfected with ΔDest-FLAG-BUGZ for 48 h using metafectene (Biontex) following the manufacturer's instructions. Cells were lysed in NP40 buffer with protease and phosphatase inhibitors, followed by sonication. Mouse anti-FLAG (Sigma-Aldrich) antibody was bound to Dynabeads Protein G (Thermo Fisher) using a ratio of 1 µl antibody/10 µl of beads prior to the addition of cell lysates. After incubation with the lysate, Dynabeads were washed 3 times with lysis buffer and proteins were eluted with SDS-Laemmli buffer and heating at 95 °C. Fluorescence tagging of PTL was performed as described above. All immunoblots and immuniprecipitations were reproduced at least 3 times.

To immunoprecipitate GFP-BUGZ, HeLa cells stably expressing GFP-BUGZ were lysed in NP40 buffer with protease and phosphatase inhibitors. ChromoTek GFP-Trap agarose beads (Proteintech) equilibrated in NP40 buffer were incubated with the cell lysate for 1 h at 4 °C. After incubation, the beads were washed 3 times with lysis buffer. An additional wash with 150 mM KCL, 2 M Urea and 0.2% SDS were done to remove all weakly bound proteins before elution using Glycine pH 2.0 followed by neutralization and digestion for mass spectrometry.

## Image processing, data visualization, and statistical analysis

All graphs and statistical analysis were generated in GraphPad Prism 8.0. Data points were tested for normality using D'Agostino & Pearson test. Accordingly, statistical significance was determined by Student's t-test (unpaired, two-tailed; normal distribution) or Mann-Whitney U-test (unpaired, two-tailed; no normal distribution). F-test was used to compare variances and Welch's correction was employed when variances were not equal. Details of the statistical significance and $n$ values for each condition can be found in the figures and figure legends. Illustrations were created in part using BioRender.com and further adjusted in Adobe Illustrator (Synopsis Figure, Appendix Fig. S2A).

## Virtual structural analysis of the ZNF207/BUGZ-parthenolide interaction

ZNF207/BUGZ is known to have two folded domains (1–110 and 270–350) with the major part of the protein being intrinsically disordered (Jiang et al, 2015). The FASTA sequence for ZNF207/BUGZ (NCBI: NP_001027464.1 ZNF207 [Homo Sapiens] [7756] [b]) was retrieved and loaded into Alfafold 3. This produced a well-organized N-terminal zinc finger structure with the two zinc binding domains. The structure was loaded into MOE (CCG) and truncated to the 117 N-terminal residues (at Asp117) by deleting the unfolded parts. The two $Zn^{2+}$ were inserted in the zinc binding cavities. The structure presented a hydrophobic pocket close to the free Cys54. The structure of Cys bound PTL was retrieved form PDB (6OCH.pdb) and was aligned with Cys54 in the ZNF207/BUGZ zinc finger, which produced a structure in which the epoxide was coordinated to $N^{\alpha}H$ and $N^{\delta}H$ of His38 with hydrogen bonds the carbonyl was coordinated to the vicinal zinc atom (2.88 Å) with only minor changes were required to provide a perfect fit of PTL. The Alfafold structure was maintained by fixing all residues not in contact (4.5 Å) with PTL. Moving atoms were soaked in a droplet of water (12) and minimizations and molecular dynamics were performed to ensure optimal interactions. When a completely stable structure was obtained (Fig. 5H; Appendix S5B,C) the Cys54-PTL-methyl bond was disconnected to give the conjugated PTL double bond. The structure was minimized and subjected to long dynamics runs showing very high stability of the non-bonded complex at 300 K (Fig. 5H; Appendix S5B,C). The stability was further tested in a box of saline for 5 ns at 300 K and 100 kPa with no constraints on any part of the molecule. The PTL binding pocket and interactions between PTL and the zinc finger did not change during this process. The Alkyne PEG-linker was attached to the methyl group of Cys54 bound PTL and the structure was soaked in a water droplet and was minimized and subjected to molecular dynamics-calculations. Even though the PEG carbonate was attached to the methyl group pointing inwards at the edge of the binding pocket, it could readily accommodate the PEG chain without loss of any important interactions. The distance between the $Zn^{2+}$ and the PTL carbonyl increase slightly from ~2.88 to ~3.20 Å.

# Data availability

The mass spectrometry data used to identify modified BUGZ peptides have been deposited to the ProteomeXchange Consortium via the PRIDE partner repository (https://www.ebi.ac.uk/pride/) with the dataset identifier PXD063178.

The source data of this paper are collected in the following database record: biostudies:S-SCDT-10_1038-S44318-025-00469-2.

## Peer review information

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

## Acknowledgements

We thank Iain Cheeseman, Jennifer DeLuca, Pavel Draber, Stephan Geley, Daniel Gerlich, Anthony Hyman, Helder Maiato, Rene Medema, and Antonio Pereira for providing reagents and tools. We thank Martina Barisic for exceptional technical assistance. We thank the Bioimaging Core Facility at the Danish Cancer Institute for support in fluorescence imaging. We thank Elena Papaleo and her team for discussions on parthenolide interactions. This work was funded by the Novo Nordisk Foundation (NNF19OC0058504, granted to MB; NNF20SA0064214, granted to MJD; and NNF22SA0081975, granted to MM), the Independent Research Fund Denmark (DFF-3101-00075, granted to MB), the Lundbeck Foundation (R434-2023-431, granted to MB; and R322-2019-2337, granted to LFG), and the Villum foundation (VF00056577, granted to MM). The Novo Nordisk Foundation Center for Protein Research is financially supported by the Novo Nordisk Foundation (no. NNF14CC0001).

## Author contributions

**Susana Eibes**: Formal analysis; Validation; Investigation; Visualization; Methodology; Writing—original draft. **R Bhagya Lakshmi**: Formal analysis; Validation; Investigation; Visualization; Methodology; Writing—review and editing. **Girish Rajendraprasad**: Formal analysis; Validation; Investigation; Visualization; Methodology; Writing—original draft; Writing—review and editing. **Brian T Weinert**: Resources; Formal analysis; Investigation; Methodology; Writing—review and editing. **Fadhil S Kamounah**: Resources; Formal analysis; Visualization; Methodology; Writing—review and editing.

**Luke F Gamon**: Formal analysis; Funding acquisition; Investigation; Visualization; Methodology; Writing—review and editing. **Sergi Rodriguez-Calado**: Formal analysis; Validation; Investigation; Visualization; Methodology; Writing—review and editing. **Morten Meldal**: Supervision; Funding acquisition; Investigation; Visualization; Methodology; Writing—review and editing. **Michael J Davies**: Resources; Supervision; Funding acquisition; Methodology; Writing—review and editing. **Michael Pittelkow**: Resources; Supervision; Funding acquisition; Methodology; Writing—review and editing. **Chunaram Choudhary**: Resources; Supervision; Funding acquisition; Methodology; Writing—review and editing. **Marin Barisic**: Conceptualization; Formal analysis; Supervision; Funding acquisition; Validation; Investigation; Visualization; Methodology; Writing—original draft; Project administration; Writing—review and editing.

Source data underlying figure panels in this paper may have individual authorship assigned. Where available, figure panel/source data authorship is listed in the following database record: biostudies:S-SCDT-10_1038-S44318-025-00469-2.

## Disclosure and competing interests statement

The authors declare no competing interests.

# Expanded View Figures

**Figure EV1.  Alkyne-parthenolide and costunolide phenocopy the mitotic effects of parthenolide.**

(**A**) Molecular structures of PTL, alkyne-PTL derivative and costunolide. (**B**) Representative confocal maximum projections of cold stable microtubules from U2OS cells treated with the indicated compounds (15 µM) and subjected to a short cold treatment. Scale bar: 10 µm. (**C**) Quantification of α-tubulin intensity at the mitotic spindle of cells treated as indicated in (**B**). N, n (N = number of cells, n = number of experiments): DMSO (57, 3), 15 µM alkyne-PTL (58, 3), 15 µM costunolide (39, 3). ****$p \leq 0.0001$. (**D**) Representative confocal max projections of mitotic U2OS cells undergoing the indicated treatments. Scale bar: 10 µm. (**E**) Quantification of astrin intensity at aligned kinetochores normalized to CENP-C intensity for DMSO, 15 µM alkyne-PTL and 15 µM costunolide treated cells. N, n (N = number of cells, n = number of experiments): DMSO (30, 3), 15 µM alkyne-PTL (30, 3), 15 µM costunolide (30, 3). ****$p \leq 0.0001$. (**F**) Immunoblot of α-tubulin detyrosination levels in U2OS cells undergoing the stated treatments. Quantification of the relative levels of detyrosination in cells undergoing the stated treatments. ****$p \leq 0.0001$, **$p \leq 0.01$. Replicates are color-coded for all quantifications. All data are presented as mean ± SD values from three independent replicates. Statistical analysis was performed using unpaired t-test with Welch's correction.

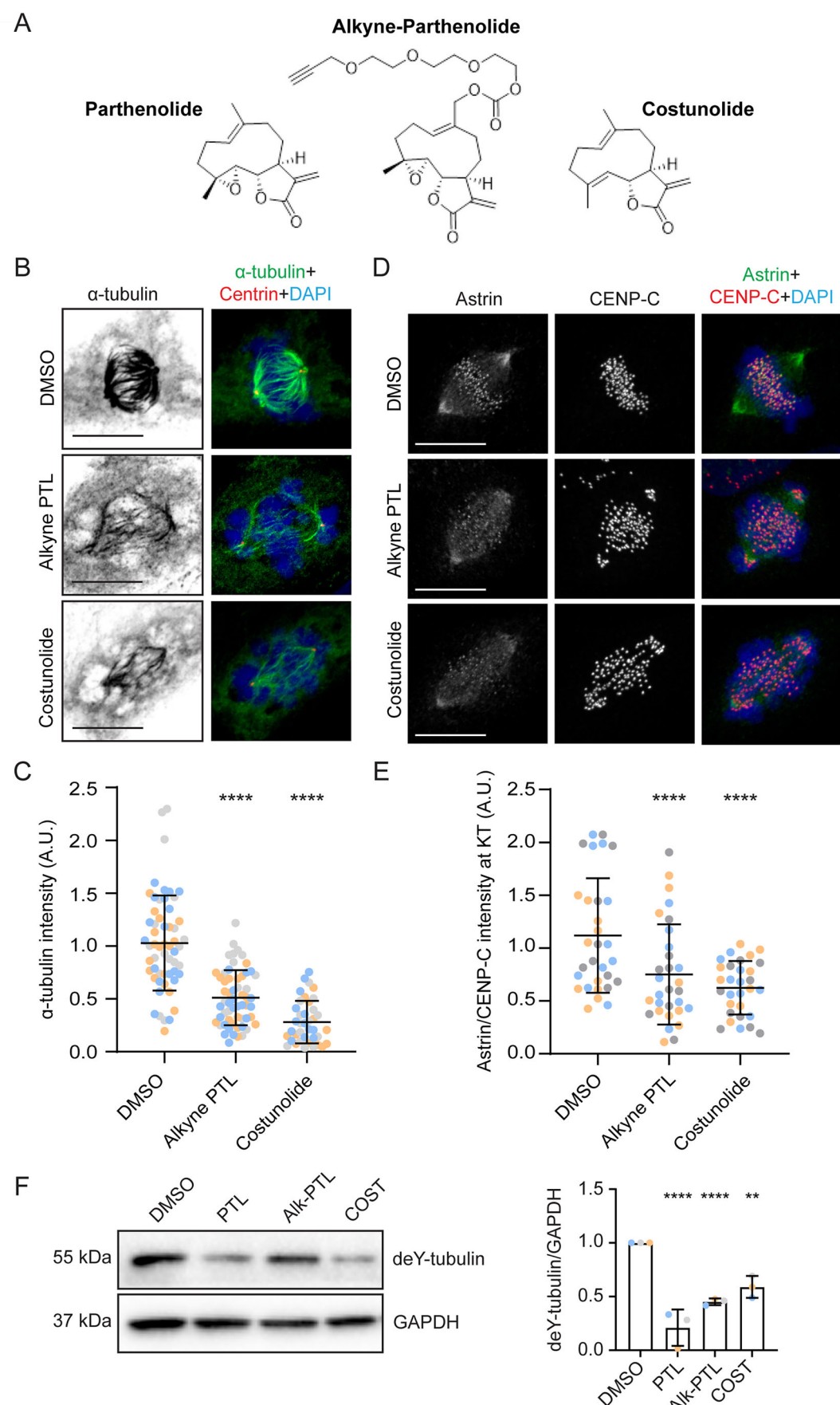

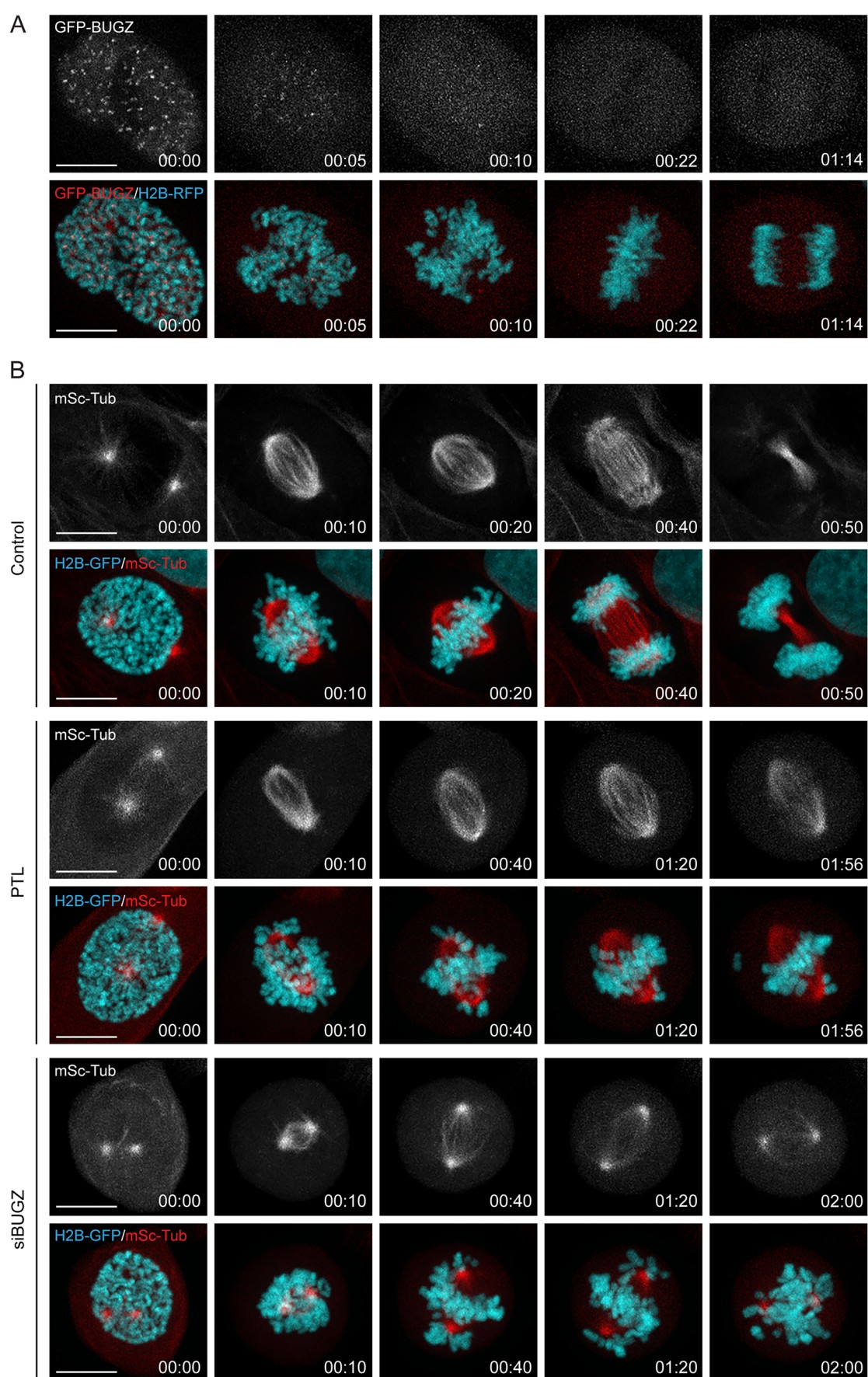

◀ **Figure EV2.  BUGZ localizes at kinetochores in early mitosis and its depletion induces mitotic defects similar to parthenolide treatment.**

(**A**) Representative spinning-disk confocal time-series of mitosis in HeLa cells stably expressing GFP-BUGZ and infected with adenovirus to express H2B-RFP. Scale bar: 10 μm. (**B**) Representative spinning-disk confocal time-series of mitosis in control, 15 μM PTL- and siBUGZ-treated U2OS cells stably expressing H2B-GFP/mScarlet-α-tubulin. Scale bar: 10 μm.

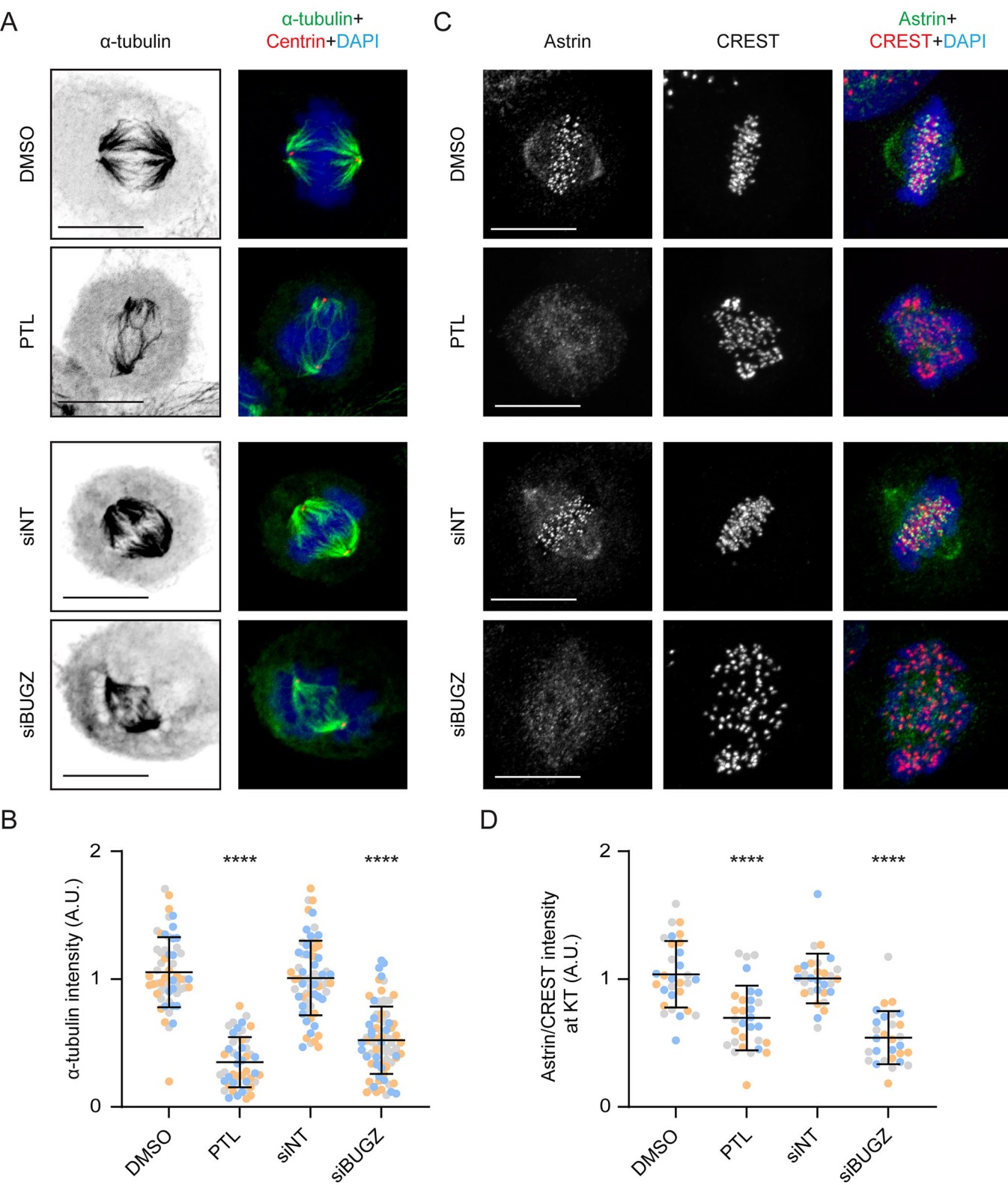

◄ **Figure EV3. BUGZ depletion disrupts kinetochore-microtubule attachments in a manner similar to parthenolide treatment.**

(A) Representative confocal max projections of cold stable microtubules from HeLa cells under the indicated conditions. Scale bar: 10 μm. (B) Quantification of α-tubulin intensity at the mitotic spindle of the cells treated as indicated in (A). *N, n* (*N* = number of cells, *n* = number of experiments): DMSO (57, 3), 15 μM PTL (49, 3), siNT (66, 3), siBUGZ (79, 3). ****$p \leq 0.0001$. (C) Representative confocal max projections of mitotic cells subjected to the indicated treatments. Scale bar: 10 μm. (D) Quantification of astrin intensity at aligned kinetochores normalized to CREST intensity for the conditions indicated in (C). *N, n* (*N* = number of cells, *n* = number of experiments): DMSO (30, 3), 15 μM PTL (30, 3), siNT (30, 3), siBUGZ (29, 3). ****$p \leq 0.0001$. Replicates are color coded for all quantifications. All data are presented as mean ± SD values from three independent replicates. Statistical analysis was performed using unpaired t-test with Welch's correction and unpaired t-test for PTL and siBUGZ, respectively, in (B). Unpaired t-test and Mann-Whitney test were used for PTL and siBUGZ, respectively, in (D).

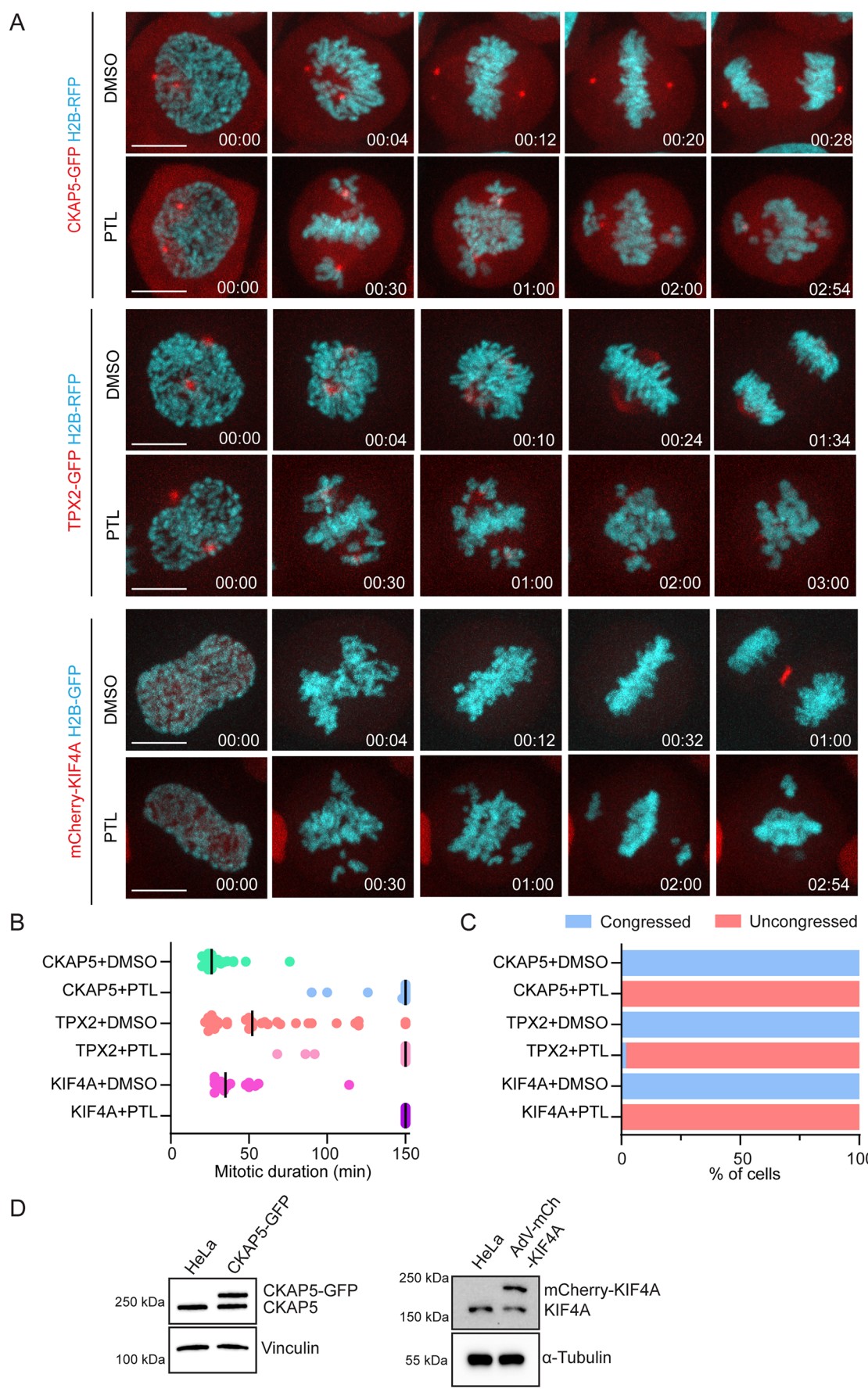

◀ **Figure EV4. Overexpression of CKAP5, TPX2 or KIF4A cannot reverse the mitotic defects induced by parthenolide.**

(A) Representative spinning-disk confocal time-series of mitosis in DMSO- and 15 µM PTL-treated HeLa cells stably expressing CKAP5-GFP, TPX2-GFP and HeLa cells transduced for expression of mCherry-KIF4A undergoing infection with adenovirus to express H2B-RFP or H2B-GFP. Scale bar: 10 µm. (B) Quantification of mitotic duration and chromosome congression status (C) in cells with indicated conditions in (A). Median is plotted for mitotic duration. $N$, $n$ ($N$ = number of cells, $n$ = number of experiments) for congression phenotype: CKAP5-GFP + DMSO (50, 3), CKAP5-GFP + 15 µM PTL (45, 3), TPX2-GFP + DMSO (50, 3), TPX2-GFP + 15 µM PTL (45, 3), mCherry-KIF4A + DMSO (26, 3), mCherry-KIF4A + 15 µM PTL (22, 3); for mitotic duration: CKAP5-GFP + DMSO (32, 3), CKAP5-GFP + 15 µM PTL (37, 3), TPX2-GFP + DMSO (32, 3), TPX2-GFP + 15 µM PTL (16, 3), mCherry-KIF4A + DMSO (16, 3), mCherry-KIF4A + 15 µM PTL (17, 3). (D) Immunoblots for cellular expression levels of CKAP5-GFP and mCherry-KIF4A.

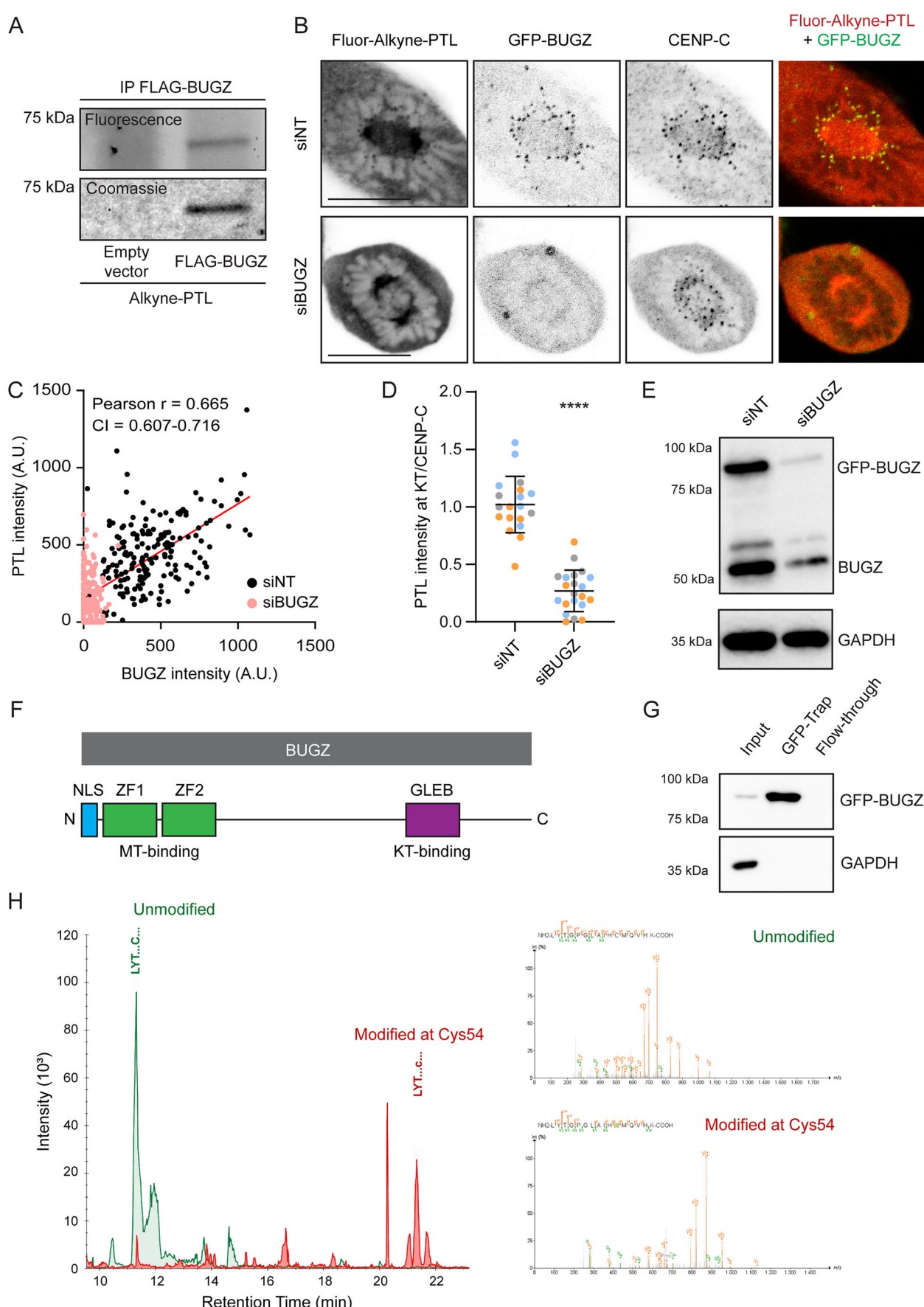

◀  **Figure EV5.  Parthenolide localizes at kinetochores by covalently binding Cys54 of BUGZ.**

(A) Fluorencence and Comassie staining of immunoprecipitated FLAG empty and FLAG-BUGZ from HEK 293T cells treated alkyne-PTL. (B) Representative confocal images of click-based imaging of 5 μM fluor-Cy5-alkyne-PTL in HeLa GFP-BUGZ cells under the indicated conditions. Scale bar: 10 μm. (C) Scatter plot showing the intensity of BUGZ (x-axis) and 5 μM fluor-Cy5-alkyne-PTL (y-axis) at individual kinetochores from the indicated conditions in (B). Each dot represents a single kinetochore. A Pearson correlation line is shown for the correlation between BUGZ and fluor-Cy5-alkyne-PTL levels. (D) Quantification of 5 μM fluor-Cy5-alkyne-PTL intensity at kinetochores normalized to CENP-C intensity for the conditions indicated in (B). N, n (number of cells, number of experiments): siNT (19, 3) siBUGZ (21, 3). ****$p \leq 0.0001$. Replicates are color coded. Data are presented as mean ± SD values from three independent replicates. Statistical analysis was performed using unpaired t-test. (E) Immunoblot for BUGZ depletion efficiency in HeLa GFP-BUGZ cells. (F) Illustration of domain architecture of BUGZ. (G) Western-blot with anti-BUGZ antibody of in cellulo GFP-Trap pulldown sample from HeLa cells stably expressing GFP-BUGZ treated with 50 μM PTL. (H) Extracted ion chromatograms and MS-MS spectra for peptides with and without PTL modification at Cys54 (red and green, respectively) from the GFP-Trap sample shown in (G).

