## [Peer Review File · The EMBO Journal]

Parthenolide disrupts mitosis by inhibiting ZNF207/BUGZ-promoted kinetochore-microtubule attachment

Susana Eibes, Bhagya R, Girish Rajendraprasad, Brian Weinert, Fadhil Kamounah, Luke Gamon, Sergi Rodriguez-Calado, Morten Meldal, Michael Davies, Michael Pittelkow, Chunaram Choudhary, and Marin Barisic

Corresponding author(s): Marin Barisic (barisic@cancer.dk)

Review Timeline:

Submission Date:	23rd Aug 24
Editorial Decision:	7th Oct 24
Revision Received:	10th Apr 25
Editorial Decision:	12th May 25
Revision Received:	14th May 25
Accepted:	15th May 25

Editor: Hartmut Vodermaier

Transaction Report:

Dr. Marin Barisic
Danish Cancer Society Research Center
Cell Division and Cytoskeleton
Strandboulevarden 49
Copenhagen 2100
Denmark

7th Oct 2024

Re: EMBOJ-2024-118847
Parthenolide disrupts mitosis by inhibiting ZNF207/BuGZ-promoted kinetochore-microtubule attachment

Dear Marin,

Thank you for submitting your study on BuGZ targeting by parthenolide to The EMBO Journal. It has now been seen by three expert referees, whose comments are copied below. As you will see, the referees appreciate the potential importance of your findings, but also raise some concerns regarding the decisiveness of the evidence in support of the proposed distinct mechanism. Should you be able to satisfactorily address these concerns, we would be open to considering a revised version of the manuscript further for publication. Given that it is our policy to allow only a single round of major revision, it would however be helpful if you contacted me with a revision plan and preliminary point-by-point response already during the early stages of your revision work, in order to clarify/discuss if and how key issues raised in the reports may be solved. We would also be open to extension of the default three-months revision period if needed; our 'scooping protection' (meaning that competing work appearing elsewhere in the meantime will not affect our considerations of your study) would of course remain valid also throughout such an extension.

Further information on preparing, formatting and uploading a revised manuscript can be found below and in our Guide to Authors. Thank you again for the opportunity to consider this work for The EMBO Journal, and I look forward to hearing from you in due time.

With kind regards,

Hartmut

4) Each main and each Expanded View (EV) figure should be uploaded as individual production-quality files (preferably in .eps, .tif, .jpg formats). For suggestions on figure preparation/layout, please refer to our Figure Preparation Guidelines:

- 5) Point-by-point response letters should include the original referee comments in full together with your detailed responses to them (and to specific editor requests if applicable), and also be uploaded as editable (e.g., .docx) text files.
- 6) Please complete our Author Checklist, and make sure that information entered into the checklist is also reflected in the manuscript; the checklist will be available to readers as part of the Review Process File. A download link is found at the top of our Guide to Authors: embopress.org/page/journal/14602075/authorguide
- 7) All authors listed as (co-)corresponding need to deposit, in their respective author profiles in our submission system, a unique ORCID identifier linked to their name. Please see our Guide to Authors for detailed instructions.
- 8) Please note that supplementary information at EMBO Press has been superseded by the 'Expanded View' for inclusion of additional figures, tables, movies or datasets; with up to five EV Figures being typeset and directly accessible in the HTML version of the article. For details and guidance, please refer to: embopress.org/page/journal/14602075/authorguide#expandedview
- 9) To facilitate reproducibility and cross-laboratory adoption of methodologies, please structure the Materials & Methods section as outlined in our guide to authors, including a completed Reagents and Tools Table that can be downloaded from our author guidelines as well (<https://www.embopress.org/page/journal/14602075/authorguide#structuredmethods>).
- 10) Digital image enhancement is acceptable practice, as long as it accurately represents the original data and conforms to community standards. If a figure has been subjected to significant electronic manipulation, this must be clearly noted in the figure legend and/or the 'Materials and Methods' section. The editors reserve the right to request original versions of figures and the original images that were used to assemble the figure. Finally, we generally encourage uploading of numerical as well as gel/blot image source data; for details see: embopress.org/page/journal/14602075/authorguide#sourcedata

At EMBO Press, we ask authors to provide source data for the main manuscript figures. Our source data coordinator will contact you to discuss which figure panels we would need source data for and will also provide you with helpful tips on how to upload and organize the files.

In the interest of ensuring the conceptual advance provided by the work, we recommend submitting a revision within 3 months (5th Jan 2025). Please discuss the revision progress ahead of this time with the editor if you require more time to complete the revisions. Use the link below to submit your revision:

Link Not Available

Referee #1:

Parthenolide is a anti-cancer agent that affects multiple cellular processes including mitosis, but its mechanism of action is not known. The drug can alter microtubule polymerization dynamics in vitro, but it is not clear whether that is the key mechanism in cells. In the present study the authors present convincing data that the major mitotic target of parthenolide is not tubulin- but rather the kinetochore protein BuGZ. They identify BuGZ through click-it chemistry/mass spec and show that the drug phenocopies the BuGZ phenotype in cells and that the drug has significantly reduced localization to kinetochores in the absence of BuGZ. They go on to identify the key site of interaction in BuGZ and use molecular modeling to show that the drug will fit within the pocket of BuGZ near the proposed site of interaction. Overall, the study is very well done with gorgeous imaging and strong and convincing data. The study is important not only because of its biomedical impact but also because it identifies a clear mechanistic target for parthenolide and also sets out a clear set of experiments of how to rigorously identify a molecular target. I am highly enthusiastic for publication of this paper with only a few small requests to clean up a few controls and fix a few minor points.

1. Technically the data in Figure 1E does show a statistically significant increase in MT nucleation in the presence of the drug. I am all in agreement about statistically significant but biologically irrelevant, but given that you look at MT intensity by imaging at 1 and 5 minutes, the corresponding MT intensity should also be measured at 5 minutes to demonstrate that while there is a small increase in nucleation at 1', by 5' the arrays are full recovered and not different between control and experimental, which is

what my eyes tell me from the images. In addition, the data in Figure 2 show that there is no effect on nucleation in mitosis, which is where the phenotype is.

2. In Figure 1- the label for part G is missing in the figure. Also, astral MTs are discussed first in the text then spindle length, but the figure is the opposite order. Fix one or the other so that they are consistent.
3. In Figure 1I, the legend does not state N, n (thank you for doing this on all of the other figures), and there are only 3 points on the figure. I am unclear as to what this data represents, but n=3 is not sufficient for half-life measurements.
4. The characterization of the derivative is insufficient. In the images, it looks much less effective in terms of the MT polymer levels in the mitotic spindle, but I agree that the movies looks similar. From Figure 2- the two quantitative measurements of the effectiveness of PTL were decreased alpha tubulin intensity and decreased astrin/Cenp-C. These need to be repeated with the costunolide and the alkyne-PTL derivatives. The discussion states that there are no obvious targets of tubulin deetyrosination so using that as a measure is not appropriate.
5. In the quantification of the effects of PTL in BugZ overexpressing cells the authors conclude that the effects are similar with 3X concentration PTL. However, the effects are not similar; even with 3X PTL, there is only 50% defect in congression, and the timing is not as severe either as only a fraction of cells arrest at 150 minutes (the duration of the experiment). The description of this data should be changed, and in the mitotic duration could you please report the percentage of cells that arrest at 2.5 hours as another comparison to control. I am curious if it is 50% of the cells like in the top part of the panel.
6. In Figure 5, the PTL intensity is only reduced by half, which could be because there are other targets or because the BugZ KD is not effective. This needs to be reconciled. While the western blot is given, there is no IF of the BugZ KD to show how much reduction there is at kinetochores.

Minor Points:

1. At the end of the first paragraph of the results, it states that the mechanism in the two cited studies has been overlooked for decades. Seems a bit dramatic given that the citations were from 2015 and 2021.
2. On pg. 10 in the middle there is a reference to Figure 3A that I think should be Figure 4A.
3. In the second paragraph on page 13, it refers to an immunoblot profile (Figure 4B), but Figure 4B is time-lapse imaging.
4. Figure 5G is not referenced in the text.

Referee #2:

In the paper entitled "Parthenolide disrupts mitosis by inhibiting ZNF207/BuGZ-promoted kinetochore-microtubule attachment" by Eibes et al., the authors describe a novel mechanism of parthenolide-induced effects on cell division. They employ a multidisciplinary approach including cell biology, chemistry, and molecular dynamics simulations to conclude that parthenolide does not affect microtubule dynamics, but instead covalently binds and inhibits BuGZ, a spindle component. I found the paper clearly written, the experiments are well-controlled and presented in a convincing manner. Particular strength of the paper is the use of a novel fluorescent analog of parthenolide that will hopefully be available to other research groups. In general, I find this paper a strong candidate for publication, and potentially a valuable addition to the field. However, there are two issues that prevent me from recommending the publication of the paper in its current form.

1. The authors state that in vivo PTL does not affect microtubule dynamics, unlike previously reported in vitro (p5/Fig 1BC). However, this conclusion is backed up by indirect evidence, and is therefore not as strong as it could be otherwise. For example, the authors measure the distance that EB-comets have travelled, and their lifetime. It seems that comet speed (a proxy for microtubule growth speed) is a much straightforward parameter that can be extracted from the same dataset. Given that the authors next report an effect of PTL on k-fiber formation (Fig 2A), and show Alkyne-PTL localization at the spindle (Fig 3C), it is important to distinguish whether the direct effect on microtubule dynamics can be completely ruled out. It would be ideal to not rely on EB as a proxy for microtubule dynamics, but instead image microtubule dynamics directly using fluorescently tagged or labelled tubulin in thin regions of spread cells.
2. The authors show convincing evidence that PTL modifies C54 of BuGZ in vitro. Is there evidence that this happens in vivo, and/or that this modification impacts MT-binding activity of BuGZ, as the authors state on p14? For example, does mutation in C54 cause the same mitotic phenotype as PTL treatment? Or, does C54 mutant of BuGZ have lower affinity for MTs in vitro?

Referee #3:

The manuscript by Eibes and colleagues seeks to examine the mechanisms of action of parthenolide (PTL) on cell division. The authors demonstrate that while PTL affects chromosome congression, it does not affect microtubule plus-end dynamics nor nucleation ability of MTOCs of cells in interphase. Also in the more dynamic state of mitosis, PTL does not seem to affect microtubule dynamics or overall spindle morphology. These findings contrast with previous suggestions that PTL directly targets microtubules. However, when assessing spindle assembly in the presence of PTL, the authors notice spindle malformation, weaker k-fiber intensities and lower astrin signals on kinetochores, all indicative of problems with establishing proper

kinetochore-microtubule interactions. To enable deeper examination, the authors then develop an alkyne-PTL analog, which retains the original activity. Visualizing alkyne PTL via click reaction, they show that alkyne-PTL does not localize to tubulin in interphase cells and localizes to kinetochores in mitotic cells. Subsequently, the authors conduct a pull-down of alkyne-PTL and perform mass spectrometry to identify potential interactants. They find several, and based on GO analysis and expected phenotype, they zoom in on BugZ, a zinc finger protein that localizes at the mitotic spindle and kinetochores and that is crucial for Bub3 loading and stabilization at the kinetochore. Overexpression of GFP-BugZ restores chromosome congression at lower PTL concentrations, and depletion of BugZ using siRNA significantly reduces alkyne-PTL localization at the kinetochore. By pinpointing the exact PTL-targeting residue (Cys54) on BugZ through LC-MS analysis on purified PTL-BugZ conjugates, they suggest that PTL disrupts the function of the second zinc finger, which is important for microtubule interactions.

The manuscript presents some compelling evidence that the main target of PTL in human cell culture is not the microtubule, as suggested previously, but that it must impact on other targets that affect chromosome congression. Those experiments are generally well executed and convincing. The authors claim a new mechanism, the inhibition of BugZ's interaction with microtubules by PTL. This part, however, needs more support. The main reasons for concluding BugZ is the main target is that 1) it is found in proteomics analyses of PTL-pulldowns. However, there are many (also more pronounced) 'hits' in the screen, and none of them are further investigated; 2) BugZ RNAi reduces fluor-alkyne-PTL localization to kinetochores. However, whether PTL localization at kinetochores is relevant for its mitotic phenotype is unclear, and moreover BugZ RNAi could have many indirect effects on kinetochores that could explain the absence of PTL. Furthermore, there are many kinetochores that retain substantial levels of PTL upon BugZ depletion (panel C). A direct correlation of BugZ levels with PTL levels at single kinetochores will be important. 3) Perhaps the most convincing piece of data on whether BugZ is the key mitotic target of PTL: BugZ overexpression prevents chromosome congression defects by PTL. However, in principle, the overexpression of any PTL-binding protein should be able to rescue the phenotype simply by lowering the PTL pool available for binding its relevant targets. This can be tested by overexpressing other targets and examining if the rescue of the mitotic phenotypes is BugZ-specific. In addition, the authors can do a lot more on examining whether the PTL phenotype correlates with a BugZ-depletion phenotype. They now discard e.g. chTOG and TPX2 as potential targets because of their published depletion phenotypes not matching the PTL phenotype but they make little effort in characterizing the BugZ-depletion phenotype with the PTL phenotype side-by-side. This not only requires a deeper analysis of the PTL phenotype (astrin levels are lower on kinetochores and cold-stable fibers are less intense, but much can be tested on the origins and consequences of this: CPC activity, MAD2-levels, SAC response, outer-kinetochore protein levels, etc) but also their own detailed analysis of BugZ-depletion phenotypes. Finally, the authors identify the residue in BugZ that binds PTL in vitro. This is very useful to show BugZ is really the target in cells. Does overexpression of a cysteine mutant prevent the rescue of the PTL phenotype? Does expression of a cysteine mutant in the background of BugZ RNAi phenocopy the PTL phenotype?

Minor comments

- The authors present colocalization between alkyne-PTL and phalloidin (actin), but in other parts of the cell (e.g. left of the nucleus in 3B) there seems to be co-localization with tubulin. This should be addressed.
- alkyne-PTL imaging was performed at 5 μ M. Most functional experiments are done at a dose of 15uM, however, so the co-stainings with alpha-tubulin, phalloidin, and CENPC should be conducted at 15 μ M also.
- The authors should perform co-staining of alkyne-PTL with BuGZ to confirm that PTL levels specifically decrease with BuGZ depletion.
- Panel 3C: why do the kinetochores light up in the α -tubulin channel?
- Figure 4: why is metaphase delayed in the BugZ overexpression experiments?
- In panel 5D, the BUB1-positive areas seem to be bigger upon PTL treatment, suggestive of more expanded fibrous coronas. This could signify that the PTL-treated cells have been in mitosis longer or that PTL affects the ability of noco to cause microtubule depolymerization. Both could impact the interpretation of the BUB1 levels
- Page 4: "BuB proteins". Also, human protein names should be written in all-caps.
- Figures and text sometimes refer to "Bugz" instead of BUGZ or BugZ
- Provide more detail about the specific statistical tests used for each figure.
- Please include the concentrations of Alkyne-PTL used in immunofluorescence imaging.
- Page 10: "Fig3A" should be "FigEV3A".

Point-by-point response to the reviewers' comments

Referee #1:

Parthenolide is a anti-cancer agent that affects multiple cellular processes including mitosis, but its mechanism of action is not known. The drug can alter microtubule polymerization dynamics in vitro, but it is not clear whether that is the key mechanism in cells. In the present study the authors present convincing data that the major mitotic target of parthenolide is not tubulin- but rather the kinetochore protein BuGZ. They identify BuGZ through click-it chemistry/mass spec and show that the drug phenocopies the BuGZ phenotype in cells and that the drug has significantly reduced localization to kinetochores in the absence of BuGZ. They go on to identify the key site of interaction in BuGZ and use molecular modeling to show that the drug will fit within the pocket of BuGZ near the proposed site of interaction. Overall, the study is very well done with gorgeous imaging and strong and convincing data. The study is important not only because of its biomedical impact but also because it identifies a clear mechanistic target for parthenolide and also sets out a clear set of experiments of how to rigorously identify a molecular target. I am highly enthusiastic for publication of this paper with only a few small requests to clean up a few controls and fix a few minor points.

We thank the reviewer for recognizing our results as important, convincing and of high quality.

1. Technically the data in Figure 1E does show a statistically significant increase in MT nucleation in the presence of the drug. I am all in agreement about statistically significant but biologically irrelevant, but given that you look at MT intensity by imaging at 1 and 5 minutes, the corresponding MT intensity should also be measured at 5 minutes to demonstrate that while there is a small increase in nucleation at 1', by 5' the arrays are full recovered and not different between control and experimental, which is what my eyes tell me from the images. In addition, the data in Figure 2 show that there is no effect on nucleation in mitosis, which is where the phenotype is.

As suggested by the reviewer, we have now also measured the microtubule intensity at t=5min in the microtubule regrowth assay, showing no statistically significant difference between the parthenolide-treated cells and controls (Fig. 1E).

2. In Figure 1- the label for part G is missing in the figure. Also, astral MTs are discussed first in the text then spindle length, but the figure is the opposite order. Fix one or the other so that they are consistent.

We have checked that all the labels are correctly placed for the corresponding figure panels in the revised manuscript. We have also fixed the order of the two graphs.

3. In Figure 1I, the legend does not state N, n (thank you for doing this on all of the other figures), and there are only 3 points on the figure. I am unclear as to what this data represents, but n=3 is not sufficient for half-life measurements.

We apologize for this omission, which we have fixed and explained in more detail in the revised version of our manuscript. The three indicated points represent half-lives extracted from the

decay curves of three independent experiments, each consisting of around 10 cells (giving around 30 cells in total per condition).

4. The characterization of the derivative is insufficient. In the images, it looks much less effective in terms of the MT polymer levels in the mitotic spindle, but I agree that the movies looks similar. From Figure 2- the two quantitative measurements of the effectiveness of PTL were decreased alpha tubulin intensity and decreased astrin/Cenp-C. These need to be repeated with the costunolide and the alkyne-PTL derivatives. The discussion states that there are no obvious targets of tubulin detyrosination so using that as a measure is not appropriate.

We thank the reviewer for this comment, which helped us to significantly improve this set of data. As suggested, we have now added spindle microtubule intensity upon cold treatment and astrin immunofluorescence quantifications for alkyne-parthenolide and customolide (Fig. EV1B-D).

5. In the quantification of the effects of PTL in BugZ overexpressing cells the authors conclude that the effects are similar with 3X concentration PTL. However, the effects are not similar; even with 3X PTL, there is only 50% defect in congression, and the timing is not as severe either as only a fraction of cells arrest at 150 minutes (the duration of the experiment). The description of this data should be changed, and in the mitotic duration could you please report the percentage of cells that arrest at 2.5 hours as another comparison to control. I am curious if it is 50% of the cells like in the top part of the panel.

We agree with the reviewer that the effect of BUGZ overexpression were slightly underestimated in the original version of our manuscript. As suggested, we have now rephrased the associated sentence into:

“a 3-fold higher concentration of PTL was required to induce similar effects in around 50% of BUGZ-overexpressing cells.”

We have also checked the mitotic duration data and found out that 36% of the cells arrested at 2.5 hours. However, this also might represent an underestimation, since some of the filmed cells have entered mitosis during the filming and thus have been scored to be arrested for less than 2.5 hours. Nevertheless, some of these cells that have been arrested for less than 2.5 hours clearly displayed strong congression problems. We have now also expanded the methods section on live-cell imaging to avoid any potential confusion with how the mitotic duration was quantified, stating:

“Mitotic duration was quantified by tracking the time spent from nuclear envelope breakdown to anaphase onset, with exception for the cells that remained in mitosis at the end of imaging, whose mitotic duration was scored as the time spent from nuclear envelope breakdown to the end of imaging.”

6. In Figure 5, the PTL intensity is only reduced by half, which could be because there are other targets or because the BugZ KD is not effective. This needs to be reconciled. While the western blot is given, there is no IF of the BugZ KD to show how much reduction there is at kinetochores.

We thank the reviewer for this important comment. Due to the lack of a good antibody for immunofluorescence detection of BUGZ, we used HeLa GFP-BUGZ cell line to compare GFP-

BUGZ with fluor-PTL in control vs. BUGZ-depleted cells (Fig. EV5B-D). In this cell line the PTL intensity at kinetochores was reduced by 75% upon BUGZ depletion. We now also show a nice correlation between the BUGZ and PTL fluorescence signals at kinetochores. With these new experiments, we confirmed BUGZ as a critical target of PTL at kinetochores. We agree with the reviewer that the remaining PTL signal at kinetochore may be due to the insufficient RNAi-based depletion, without excluding the possibility of additional binding to some other kinetochore target.

Minor Points:

1. At the end of the first paragraph of the results, it states that the mechanism in the two cited studies has been overlooked for decades. Seems a bit dramatic given that the citations were from 2015 and 2021.

We would like to clarify that we state that the phenotype (reported in 2015), not the mechanism, was overlooked for decades.

2. On pg. 10 in the middle there is a reference to Figure 3A that I think should be Figure 4A.

We would like to thank the reviewer for spotting this mistake. We have corrected this now into Appendix Fig. S3A.

3. In the second paragraph on page 13, it refers to an immunoblot profile (Figure 4B), but Figure 4B is time-lapse imaging.

We have fixed this now by also referring to Appendix Fig. S3E.

4. Figure 5G is not referenced in the text.

Thank you for drawing our attention to this omission, which we have now corrected.

Referee #2:

In the paper entitled "Parthenolide disrupts mitosis by inhibiting ZNF207/BuGZ-promoted kinetochore-microtubule attachment" by Eibes et al., the authors describe a novel mechanism of parthenolide-induced effects on cell division. They employ a multidisciplinary approach including cell biology, chemistry, and molecular dynamics simulations to conclude that parthenolide does not affect microtubule dynamics, but instead covalently binds and inhibits BuGZ, a spindle component. I found the paper clearly written, the experiments are well-controlled and presented in a convincing manner. Particular strength of the paper is the use of a novel fluorescent analog of parthenolide that will hopefully be available to other research groups. In general, I find this paper a strong candidate for publication, and potentially a valuable addition to the field. However, there are two issues that prevent me from recommending the publication of the paper in its current form.

We thank the reviewer for recognizing our results as convincing, well-controlled and valuable for the field.

1. The authors state that in vivo PTL does not affect microtubule dynamics, unlike previously reported in vitro (p5/Fig 1BC). However, this conclusion is backed up by indirect evidence, and is therefore not as strong as it could be otherwise. For example, the authors measure the distance that EB-comets have travelled, and their lifetime. It seems that comet speed (a proxy for microtubule growth speed) is a much straightforward parameter that can be extracted from the same dataset. Given that the authors next report an effect of PTL on k-fiber formation (Fig 2A), and show Alkyne-PTL localization at the spindle (Fig 3C), it is important to distinguish whether the direct effect on microtubule dynamics can be completely ruled out. It would be ideal to not rely on EB as a proxy for microtubule dynamics, but instead image microtubule dynamics directly using fluorescently tagged or labelled tubulin in thin regions of spread cells.

We thank the reviewer for these useful suggestions on how to address and present the eventual direct effect of parthenolide on microtubule dynamics. As suggested by the reviewer, we have now added quantification of EB1-comet speed, as a proxy for the microtubule growth speed, showing no effect of PTL (Appendix Fig. S1B). We would also like to comment that we have not used EB1 comets as a single assay to address the effect of parthenolide on microtubule dynamics. We have also assessed this by using microtubule regrowth assay from centrosomes, both in interphase and mitosis, as well as by using immunofluorescence of astral microtubule formation in mitosis. In addition, in order to measure the microtubule dynamics more directly, we used photoactivation of PA-GFP-alpha tubulin in living cells and analyzed microtubule turnover of both “fast” and “slow” fractions of mitotic spindle microtubules (resembling interpolar and kinetochore microtubules, respectively).

2. The authors show convincing evidence that PTL modifies C54 of BuGZ in vitro. Is there evidence that this happens in vivo, and/or that this modification impacts MT-binding activity of BuGZ, as the authors state on p14? For example, does mutation in C54 cause the same mitotic phenotype as PTL treatment? Or, does C54 mutant of BuGZ have lower affinity for MTs in vitro?

We thank the reviewer for these important comments. To address whether parthenolide modifies C54 of BUGZ in vivo, we have pulled down GFP-BUGZ from parthenolide-treated HeLa GFP-BUGZ cells and analyzed the sample using mass-spectrometry (Fig. EV5G,H). Similar to our earlier in vitro data, we were able to detect the modified C54 of BUGZ by this approach too.

To assess the phenotype of BUGZ C54 mutant and its ability to bind parthenolide, we have used lentivirus-based technology to generate stable cell lines expressing GFP-tagged WT and C54A mutant. We then monitored the behaviour of these cell lines by live-cell imaging of GFP-BUGZ and adenoviral expression of H2B-RFP. First, we observed that we could not achieve similar expression levels of the mutants compared to the WT BUGZ. Also, around 50% of the mutant-expressing cells have shown chromosome congression defects similar to BUGZ depletion or PTL treatment (please see Figures R1 and R2 below). We conclude that the mutations interfered with the contribution of the ZNF domain of BUGZ for its mitotic function, which led to the mitotic phenotype and inability to grow the cells with high expression of the mutants.

Referee #3:

The manuscript by Eibes and colleagues seeks to examine the mechanisms of action of parthenolide (PTL) on cell division. The authors demonstrate that while PTL affects chromosome congression, it does not affect microtubule plus-end dynamics nor nucleation ability of MTOCs of cells in interphase. Also in the more dynamic state of mitosis, PTL does not seem to affect microtubule dynamics or overall spindle morphology. These findings contrast with previous suggestions that PTL directly targets microtubules. However, when assessing spindle assembly in the presence of PTL, the authors notice spindle malformation, weaker k-fiber intensities and lower astrin signals on kinetochores, all indicative of problems with establishing proper kinetochore-microtubule interactions. To enable deeper examination, the authors then develop an alkyne-PTL analog, which retains the original activity. Visualizing alkyne PTL via click reaction, they show that alkyne-PTL does not localize to tubulin in interphase cells and localizes to kinetochores in mitotic cells. Subsequently, the authors conduct a pull-down of alkyne-PTL and perform mass spectrometry to identify potential interactants. They find several, and based on GO analysis and expected phenotype, they zoom in on BugZ, a zinc finger protein that localizes at the mitotic spindle and kinetochores and that is crucial for Bub3 loading and stabilization at the kinetochore. Overexpression of GFP-BugZ restores chromosome congression at lower PTL concentrations, and depletion of BugZ using siRNA significantly reduces alkyne-PTL localization at the kinetochore. By pinpointing the exact PTL-targeting residue (Cys54) on BugZ through LC-MS analysis on purified PTL-BugZ conjugates, they suggest that PTL disrupts the function of the second zinc finger, which is important for microtubule interactions.

The manuscript presents some compelling evidence that the main target of PTL in human cell culture is not the microtubule, as suggested previously, but that it must impact on other targets that affect chromosome congression. Those experiments are generally well executed and convincing. The authors claim a new mechanism, the inhibition of BugZ's interaction with microtubules by PTL. This part, however, needs more support.

We thank the reviewer for recognizing our experiments as well-executed and convincing, but also for providing important constructive criticism required to improve our conclusions.

The main reasons for concluding BugZ is the main target is that 1) it is found in proteomics analyses of PTL-pulldowns. However, there are many (also more pronounced) 'hits' in the screen, and none of them are further investigated;

Compared to the other candidates, we initially selected BUGZ as the best-fitting one due to its mitotic localization, function and depletion-based phenotype. As suggested by the reviewer, in our revised manuscript, we have expanded the comparison between the BUGZ depletion and parthenolide effect. We have also compared the effect of BUGZ overexpression on parthenolide efficacy with the overexpression of other three candidates (CKAP5/chTOG, TPX2 and KIF4A). Please see below for more detail.

2) BugZ RNAi reduces fluor-alkyne-PTL localization to kinetochores. However, whether PTL localization at kinetochores is relevant for its mitotic phenotype is unclear, and moreover BugZ RNAi could have many indirect effects on kinetochores that could explain the absence of PTL. Furthermore, there are many kinetochores that retain substantial levels of PTL upon BugZ depletion (panel C). A direct correlation of BugZ levels with PTL levels at single kinetochores will be important.

We thank the reviewer for this important comment, which allowed us to assess the effect of BUGZ depletion on parthenolide localization at kinetochores in a more accurate way. Due to the lack of a good antibody for immunofluorescence detection of BUGZ, we used HeLa GFP-BUGZ cell line to compare GFP-BUGZ with fluor-PTL in control vs. BugZ-depleted cells (Fig. EV5B-D). In this cell line, the PTL intensity at kinetochores was reduced by 75% upon BUGZ depletion. We have also performed a direct correlation of BUGZ levels with PTL levels at single kinetochores and showed a nice correlation between the BUGZ and PTL fluorescence signals at kinetochores. We believe that the remaining PTL signal at kinetochore may result from insufficient RNAi-based depletion, without excluding the possibility of additional binding to some other kinetochore target.

3) Perhaps the most convincing piece of data on whether BugZ is the key mitotic target of PTL: BugZ overexpression prevents chromosome congression defects by PTL. However, in principle, the overexpression of any PTL-binding protein should be able to rescue the phenotype simply by lowering the PTL pool available for binding its relevant targets. This can be tested by overexpressing other targets and examining if the rescue of the mitotic phenotypes is BugZ-specific.

We thank the reviewer for raising this important concern. To test whether the overexpression of BUGZ or any other PTL-binding protein simply sequester PTL and thus rescue the phenotype by preventing its binding to other relevant targets, we compared BUGZ overexpression with equivalent overexpression of three other PTL-binding mitotic proteins: CKAP5/chTOG, TPX2 and KIF4A, and assessed their effect on PTL-induced mitotic problems. We show that the overexpression of none of these three additional proteins was able to rescue the PTL-associated mitotic phenotypes, indicating that the rescue observed in HeLa GFP-BUGZ cells was BUGZ-specific (Fig. EV4A-D).

In addition, the authors can do a lot more on examining whether the PTL phenotype correlates with a BugZ-depletion phenotype. They now discard e.g. chTOG and TPX2 as potential targets because of their published depletion phenotypes not matching the PTL phenotype but they make little effort in characterizing the BugZ-depletion phenotype with the PTL phenotype side-by-side. This not only requires a deeper analysis of the PTL phenotype (astrin levels are lower on kinetochores and cold-stable fibers are less intense, but much can be tested on the origins and consequences of this: CPC activity, MAD2-levels, SAC response, outer-kinetochore protein levels, etc) but also their own detailed analysis of BugZ-depletion phenotypes.

We agree with the reviewer that a more rigorous comparison of BUGZ depletion phenotypes with parthenolide-induced effects would strengthen the association between parthenolide and BUGZ as its proposed target. We have now expanded our analysis by quantifying the effects of BUGZ depletion on microtubule stability after cold treatment, as well as on kinetochore localization of astrin (Fig. EV3A-D). We also provide a comparison of the effects of BUGZ depletion and parthenolide treatment on kinetochore localization of MAD2 (Appendix Fig. S4).

Finally, the authors identify the residue in BugZ that binds PTL in vitro. This is very useful to show BugZ is really the target in cells. Does overexpression of a cysteine mutant prevent the rescue of the PTL phenotype? Does expression of a cysteine mutant in the background of BugZ RNAi phenocopy the PTL phenotype?

We thank the reviewer for this important comment. To assess the phenotype of BUGZ C54 mutant and its ability to bind parthenolide, we generated stable cell lines expressing GFP-

tagged WT and C54A mutant. We then monitored the behaviour of these cell lines by live-cell imaging of GFP-BUGZ and adenoviral expression of H2B-RFP. First, we observed that we could not achieve similar expression levels of the mutants compared to the WT BUGZ. Also, around 50% of RNAi-resistant mutant-expressing cells were not able to rescue the depletion of endogenous BUGZ, showing chromosome congression defects similar to BUGZ depletion or PTL treatment (please see Figures R1 and R2 below). We conclude that the mutations have most likely interfered with the contribution of the ZNF domain of BUGZ for its mitotic function, which led to the mitotic phenotype and inability to grow the cells with high expression of the mutants.

Minor comments

- The authors present colocalization between alkyne-PTL and phalloidin (actin), but in other parts of the cell (e.g. left of the nucleus in 3B) there seems to be co-localization with tubulin. This should be addressed.

We have re-checked our interphase stainings and could not detect any colocalization between alkyne-PTL and microtubules. We trust that alkyne-PTL colocalizes with actin, not microtubules in the area left of the nucleus in Fig. 3B.

- alkyne-PTL imaging was performed at 5 μ M. Most functional experiments are done at a dose of 15 μ M, however, so the co-stainings with alpha-tubulin, phalloidin, and CENPC should be conducted at 15 μ M also.

We thank the reviewer for this comment, which allowed us to correct any confusion that might have been present regarding the alkyne-PTL concentrations. We only opted for 5 μ M in two occasions: data presented in Appendix Fig. S2B and Fig. EV5B-D. In Appendix Fig. S2B we used 5 μ M alkyne-PTL to weaken the PTL-induced phenotype and allow cells to progress through mitosis, which allowed us to follow the alkyne-PTL localization through all mitotic phases. In Fig. EV5B-D we used 5 μ M alkyne-PTL to reduce the background associated with Fluor Cy5 azide, which we needed to use instead of Fluor 488 to co-stain alkyne-PTL with GFP-BUGZ.

Everything else was imaged using 15 μ M alkyne-PTL, including all other co-stainings with alpha-tubulin, phalloidin, CENPC and CREST. This is now clearly stated throughout the manuscript.

- The authors should perform co-staining of alkyne-PTL with BuGZ to confirm that PTL levels specifically decrease with BuGZ depletion.

As discussed above, due to the lack of a good antibody for immunofluorescence detection of BUGZ, we used HeLa GFP-BUGZ cell line to compare GFP-BUGZ with fluor-PTL in control vs. BUGZ-depleted cells (Fig. EV5B-D). In this cell line, the PTL intensity at kinetochores was reduced by 75% upon BUGZ depletion. We have also performed a direct correlation of BUGZ levels with PTL levels at single kinetochores and showed a nice correlation between the BUGZ and PTL fluorescence signals at kinetochores.

- Panel 3C: why do the kinetochores light up in the α -tubulin channel?

We thank the reviewer for spotting this crosstalk between kinetochore and tubulin channels. We have now replaced Fig. 3C with a more representative image.

- Figure 4: why is metaphase delayed in the BugZ overexpression experiments?

We reason that the slight delay in metaphase duration in BUGZ overexpressing cells is due to BUGZ's function in loading BUB1 and BUB3 to kinetochores. This is regulated by the GLEBS domain of BUGZ and is independent of its ZNF domain.

- In panel 5D, the BUB1-positive areas seem to be bigger upon PTL treatment, suggestive of more expanded fibrous coronas. This could signify that the PTL-treated cells have been in mitosis longer or that PTL affects the ability of noco to cause microtubule depolymerization. Both could impact the interpretation of the BUB1 levels

We thank the reviewer for drawing our attention to this issue. We have looked carefully into the rest of BUB1 stainings and realized that there was no difference in the appearance of BUB1-positive areas, which is supported by unchanged BUB1 intensity at kinetochores quantified in Fig. 5E. Therefore, to avoid any confusion, we have now replaced Fig. 5D with a more representative panel.

- Page 4: "BuB proteins". Also, human protein names should be written in all-caps.

We have fixed this.

- Figures and text sometimes refer to "Bugz" instead of BUGZ or BugZ

We have fixed this.

- Provide more detail about the specific statistical tests used for each figure.

In addition to detailed description in the methods section, we now also provide more detail in the figure legends on specific statistical tests used in each figure.

- Please include the concentrations of Alkyne-PTL used in immunofluorescence imaging.

We have fixed this.

- Page 10: "Fig3A" should be "FigEV3A".

We have fixed this.

FIGURES

Figure R1

Figure R1. A) Representative spinning disk confocal time-series of mitosis in HeLa cells stably expressing GFP-BUGZ wildtype (WT) or C54A mutant and infected with adenovirus for expression of H2B-RFP. Scale bar: 10 μ m. Immunoblot with anti-BUGZ antibody of protein extract from HeLa cells stably expressing GFP-BUGZ WT or mutant.

Figure R2

Figure R2. A) Representative spinning disk confocal time-series of mitosis in HeLa cells stably expressing RNAi-resistant GFP-BUGZ wildtype or C54A mutant, infected with adenovirus for expression of H2B-RFP and undergoing the indicated conditions. Scale bar: 10 μ m. B) Quantification of chromosome congression status in cells with the conditions indicated in A. N, n (N = number of cells, n = number of experiments): GFP-BUGZ WT + DMSO (57, 3), GFP-BUGZ WT + 15 μ M PTL (97, 3), GFP-BUGZ C54A + DMSO (61, 3), GFP-BUGZ C54A + 15 μ M PTL (46, 3), GFP-BUGZ WT + siBUGZ (102, 3), GFP-BUGZ WT + siBUGZ + 15 μ M PTL (68, 3), GFP-BUGZ C54A + siBUGZ (34, 3), GFP-BUGZ C54A + siBUGZ + 15 μ M PTL (28, 3).

Dr. Marin Barisic
Danish Cancer Institute
Cell Division and Cytoskeleton
Strandboulevarden 49
Copenhagen 2100
Denmark

12th May 2025

Re: EMBOJ-2024-118847R
Parthenolide disrupts mitosis by inhibiting ZNF207/BUGZ-promoted kinetochore-microtubule attachment

Dear Marin,

Thank you for submitting your revised manuscript for our consideration. Two of the original reviewers have now looked at it once more, and I am pleased to say that they were broadly satisfied with your responses and revisions. After incorporation of minor textual modifications requested by the referees, we shall therefore be happy to accept your manuscript for EMBO Journal publication.

In addition, please make sure to also address the following remaining editorial issues:

- On the abstract page of the manuscript, please include 4-5 general keyword terms to enhance searchability.
- Please carefully go through the reference list and make sure that each reference is complete with citation year, volume, and page/locator numbers.
- Please rename the Conflict of Interest section into "Disclosure and Competing Interests Statement", in accordance with our updated Guide to Authors (<https://www.embopress.org/competing-interests>)
- As we are switching from a free-text author contribution statement towards a more formal statement based on Contributor Role Taxonomy (CRediT) terms, please remove the present Author Contribution section and instead specify each author's contribution(s) directly in the Author Information page of our submission system during upload of the final manuscript. See <https://casrai.org/credit/> for more information.
- In the Data Availability section, please add a URL linking to the database in which the listed data from the manuscript has been deposited.
- Please remove the Appendix figure legends from the main manuscript file, they should only appear in the Appendix itself.
- In the Appendix, please remove the figure numbers (S6-S22) from the figure provided in the Appendix Material and Method section. If you would like to retain the figure numbers, they would need to be added to the Appendix Table of Contents, and each of them referenced from the text at least once.
- Finally, please provide suggestions for a short 'blurb' text prefacing and summing up the conceptual aspect of the study in two sentences (max. 250 characters), followed by 3-5 one-sentence 'bullet points' with brief factual statements of key results of the paper; they will form the basis of an editor-written 'Synopsis' accompanying the online version of the article. Please also upload a synopsis image, which can be used as a "visual title" for the synopsis section of your paper. The image should be in PNG or JPG format, and please make sure that it remains in the modest dimensions of (EXACTLY) 550 PIXELS WIDE and 300-600 PIXELS HIGH.

I am therefore returning the manuscript to you for a final round of revision, to allow you to make these modifications and upload the revised files. Once we will have received them, we should be ready to swiftly proceed with formal acceptance and production of the manuscript.

With kind regards,

Hartmut

Hartmut Vodermaier, PhD
Senior Editor, The EMBO Journal

- 1) Every manuscript requires a Data Availability section (even if only stating that no deposited datasets are included). Primary datasets or computer code produced in the current study have to be deposited in appropriate public repositories prior to resubmission, and reviewer access details provided in case that public access is not yet allowed. Further information: embopress.org/page/journal/14602075/authorguide#dataavailability
- 2) Each figure legend must specify
 - size of the scale bars that are mandatory for all micrograph panels
 - the statistical test used to generate error bars and P-values
 - the type error bars (e.g., S.E.M., S.D.)
 - the number (n) and nature (biological or technical replicate) of independent experiments underlying each data point
 - Figures may not include error bars for experiments with $n < 3$; scatter plots showing individual data points should be used instead.
- 3) Revised manuscript text (including main tables, and figure legends for main and EV figures) has to be submitted as editable text file (e.g., .docx format). We encourage highlighting of changes (e.g., via text color) for the referees' reference.
- 4) Each main and each Expanded View (EV) figure should be uploaded as individual production-quality files (preferably in .eps, .tif, .jpg formats). For suggestions on figure preparation/layout, please refer to our Figure Preparation Guidelines: <http://bit.ly/EMBOPressFigurePreparationGuideline>
- 5) Point-by-point response letters should include the original referee comments in full together with your detailed responses to them (and to specific editor requests if applicable), and also be uploaded as editable (e.g., .docx) text files.
- 6) Please complete our Author Checklist, and make sure that information entered into the checklist is also reflected in the manuscript; the checklist will be available to readers as part of the Review Process File. A download link is found at the top of our Guide to Authors: embopress.org/page/journal/14602075/authorguide
- 7) All authors listed as (co-)corresponding need to deposit, in their respective author profiles in our submission system, a unique ORCID identifier linked to their name. Please see our Guide to Authors for detailed instructions.
- 8) Please note that supplementary information at EMBO Press has been superseded by the 'Expanded View' for inclusion of additional figures, tables, movies or datasets; with up to five EV Figures being typeset and directly accessible in the HTML version of the article. For details and guidance, please refer to: embopress.org/page/journal/14602075/authorguide#expandedview
- 9) To facilitate reproducibility and cross-laboratory adoption of methodologies, please structure the Materials & Methods section as outlined in our guide to authors, including a completed Reagents and Tools Table that can be downloaded from our author guidelines as well (<https://www.embopress.org/page/journal/14602075/authorguide#structuredmethods>).
- 10) Digital image enhancement is acceptable practice, as long as it accurately represents the original data and conforms to community standards. If a figure has been subjected to significant electronic manipulation, this must be clearly noted in the figure legend and/or the 'Materials and Methods' section. The editors reserve the right to request original versions of figures and the original images that were used to assemble the figure. Finally, we generally encourage uploading of numerical as well as gel/blot image source data; for details see: embopress.org/page/journal/14602075/authorguide#sourcedata

In the interest of ensuring the conceptual advance provided by the work, we recommend submitting a revision within 3 months (10th Aug 2025). Please discuss the revision progress ahead of this time with the editor if you require more time to complete the revisions. Use the link below to submit your revision:

Link Not Available

Referee #1:

In this revision the authors have done an outstanding job of addressing the reviewer concerns. The paper was already strong, and the additional controls/data make it even stronger. I am highly enthusiastic about publication of this work. Just one minor typo- on the bottom of page 5 the authors need to include the units in the text for the EB1 comet velocity.

Referee #3:

the authors have done a good job of addressing my comments. I support publication of this interesting manuscript. I would urge the authors, however, to be more explicit in their results/discussion about two things:

1: Fig EV5C is an informative experiment but in fact shows that there can be very substantial PTL at kinetochores that show no detectable BUGZ. This means that there are very likely one or more kinetochore proteins binding PTL, to quite a substantial degree. I think this needs to be pointed out more explicitly

2: The C54 mutant indeed cannot be used to probe the whether the mechanism of PTL effects in cells is through binding BUGZ as the mutant by itself is defective in function. That is important information for readers to have, both for addressing an obvious lacuna in the data and for future experiments others might think of doing. I would urge the authors to put the rebuttal fig in the supplement and mention it in the results or discussion. Or at the very least use a 'data not shown' remark.

May 14, 2025

Danish Cancer Society

Dr. Hartmut Vodermaier
Senior Editor
The EMBO Journal

Marin Barisic
Research Group Leader
Strandboulevarden 49
DK-2100 København Ø
Denmark

Tlf +45 4535 25 73 23

Mob +45 4528 99 01 64

barisic@cancer.dk
www.cancer.dk

Dear Hartmut,

Please find enclosed our revised manuscript along with the changes recommended by the reviewers. Here is a detailed description of the changes:

“Referee #1:

In this revision the authors have done an outstanding job of addressing the reviewer concerns. The paper was already strong, and the additional controls/data make it even stronger. I am highly enthusiastic about publication of this work. Just one minor typo- on the bottom of page 5 the authors need to include the units in the text for the EB1 comet velocity.”

- We have included the units for the EB1 comet velocity on the bottom of the page 5.

“Referee #3:

the authors have done a good job of addressing my comments. I support publication of this interesting manuscript. I would urge the authors, however, to be more explicit in their results/discussion about two things:

1: Fig EV5C is an informative experiment but in fact shows that there can be very substantial PTL at kinetochores that show no detectable BUGZ. This means that there are very likely one or more kinetochore proteins binding PTL, to quite a substantial degree. I think this needs to be pointed out more explicitly

2: The C54 mutant indeed cannot be used to probe the whether the mechanism of PTL effects in cells is through binding BUGZ as the mutant by itself is defective in function. That is important information for readers to have, both for addressing an obvious lacuna in the data and for future experiments others might think of doing. I would urge the authors to put the rebuttal fig in the supplement and mention it in the results or discussion. Or at the very least use a 'data not shown' remark.”

- Regarding the data associated with Fig. EV5C, we have added the following text at the bottom of page 14:

“The remaining PTL signal at kinetochores may result from additional binding to other kinetochore target, such as CKAP5/chTOG identified in this study, and/or from insufficient RNAi-based depletion.”

- We have included the C54A mutant data into Appendix Figs. S6 and S7, described the experiments on pages 16-17, added a methods section describing the generation of the mutant, and updated the reagents table accordingly.

Kind regards,

Marin Barisic (on behalf of all authors)

Dr. Marin Barisic
Danish Cancer Institute
Cell Division and Cytoskeleton
Strandboulevarden 49
Copenhagen 2100
Denmark

15th May 2025

Re: EMBOJ-2024-118847R1
Parthenolide disrupts mitosis by inhibiting ZNF207/BUGZ-promoted kinetochore-microtubule attachment

Dear Marin,

Thank you for submitting your final revised manuscript for our consideration. I am pleased to inform you that we have now accepted it for publication in The EMBO Journal.

With kind regards,

Hartmut
